# 16p11.2 microdeletion imparts transcriptional alterations in human iPSC-derived models of early neural development

Julien G Roth[1†], Kristin L Muench[1†], Aditya Asokan[1], Victoria M Mallett[1], Hui Gai[1,2], Yogendra Verma[1], Stephen Weber[1], Carol Charlton[1], Jonas L Fowler[1], Kyle M Loh[1], Ricardo E Dolmetsch[2‡], Theo D Palmer[1*]

[1]Department of Neurosurgery and The Institute for Stem Cell Biology and Regenerative Medicine, Stanford University School of Medicine, Stanford, United States; [2]Department of Neurobiology, Stanford University School of Medicine, Stanford, United States

**\*For correspondence:**
tpalmer@stanford.edu

[†]These authors contributed equally to this work

**Present address:** [‡]Department of Neuroscience, Novartis Institutes for BioMedical Research, Cambridge, United States

**Competing interests:** The authors declare that no competing interests exist.

**Abstract** Microdeletions and microduplications of the 16p11.2 chromosomal locus are associated with syndromic neurodevelopmental disorders and reciprocal physiological conditions such as macro/microcephaly and high/low body mass index. To facilitate cellular and molecular investigations into these phenotypes, 65 clones of human induced pluripotent stem cells (hiPSCs) were generated from 13 individuals with 16p11.2 copy number variations (CNVs). To ensure these cell lines were suitable for downstream mechanistic investigations, a customizable bioinformatic strategy for the detection of random integration and expression of reprogramming vectors was developed and leveraged towards identifying a subset of 'footprint'-free hiPSC clones. Transcriptomic profiling of cortical neural progenitor cells derived from these hiPSCs identified alterations in gene expression patterns which precede morphological abnormalities reported at later neurodevelopmental stages. Interpreting clinical information—available with the cell lines by request from the Simons Foundation Autism Research Initiative—with this transcriptional data revealed disruptions in gene programs related to both nervous system function and cellular metabolism. As demonstrated by these analyses, this publicly available resource has the potential to serve as a powerful medium for probing the etiology of developmental disorders associated with 16p11.2 CNVs.

## Introduction

Copy number variations (CNVs) play a role in the etiology of various neuropsychiatric disorders including intellectual disability (*Malhotra and Sebat, 2012*), developmental delay (*Malhotra and Sebat, 2012*), congenital malformations (*Cooper et al., 2011*), autism spectrum disorder (ASD) (*Sebat et al., 2007*; *Pinto et al., 2010*), schizophrenia (SCZ) (*Walsh et al., 2008*; *Levinson et al., 2011*), bipolar disorder (BD) (*Rucker et al., 2013*), and recurrent depression (*Malhotra et al., 2011*). Microdeletions and microduplications of a 593 kb region of chromosome 16p11.2 have been implicated as a penetrant risk factor in the onset of neurodevelopmental disorders including ASD and SCZ (*Weiss et al., 2008*; *McCarthy et al., 2009*). This chromosomal region spans 29.4–32.2 Mb in the reference genome (GRCh37/hg19) and encompasses 29 genes, of which 25 are protein coding (*Weiss et al., 2008*; *Jacquemont et al., 2011*; *Blumenthal et al., 2014*). While estimates of prevalence vary, multiple studies report the presence of a deletion or duplication at the 16p11.2 chromosomal locus in approximately 1% of individuals with ASD (*Sebat et al., 2007*; *Weiss et al., 2008*;

*Jacquemont et al., 2011*; *Stefansson et al., 2014*) and between 0.01% and 0.1% of the general population (*Weiss et al., 2008*; *Jacquemont et al., 2011*; *Stefansson et al., 2014*). A meta-analysis of seven studies suggests an overall prevalence of 0.76% 16p11.2 CNVs among idiopathic ASD probands (*Walsh and Bracken, 2011*).

The behavioral and physiological phenotypes associated with the 16p11.2 CNV include reciprocal, shared, and gene dosage-dependent abnormalities [*Figure 1A*]. Both microdeletions and microduplications of 16p11.2 are associated with ASD, although ASD represents a greater proportion of the diagnoses associated with 16p11.2 deletion (*Walsh and Bracken, 2011*; *Glessner et al., 2009*; *Shen et al., 2010*). Conversely, the risk of SCZ is greater in 16p11.2 microduplication carriers (*Walsh et al., 2008*; *McCarthy et al., 2009*). Reciprocal neuroanatomical phenotypes of the 16p11.2 CNV include differences in head size and brain volume. Specifically, individuals with a microdeletion of the 16p11.2 locus present with macrocephaly, increased overall gray and white matter volumes, increased cortical surface area, and increased axial diffusivity of white matter tracts such as the anterior corpus callosum and the internal and external capsules (*Shinawi et al., 2010*; *Qureshi et al., 2014*; *Owen et al., 2014*). Individuals with a microduplication of 16p11.2 present with microcephaly and corresponding decreases in gray matter, white matter, and cortical surface area (*Shinawi et al., 2010*; *Qureshi et al., 2014*). Recent studies have characterized independent and reciprocal abnormalities in both auditory processing delays and the amplitude of visual evoked potentials among individuals with the 16p11.2 CNVs (*Jenkins et al., 2016*; *LeBlanc and Nelson, 2016*). Finally, obesity and hyperphagia are observed among deletion carriers, while low body mass index (BMI) is observed in duplication carriers (*Jacquemont et al., 2011*; *Walters et al., 2010*).

The underlying cellular and molecular mechanisms by which neuropsychiatric disorders develop remain largely unknown. Investigators probing the etiology of neurodevelopmental disorders acquired a powerful new tool with the discovery that human somatic cells can be reprogrammed into human induced pluripotent stem cells (hiPSCs) which, in turn, can be differentiated into ectodermal, endodermal, and mesodermal derivatives (*Takahashi et al., 2007*; *Yu et al., 2007*). In the years that followed, the field of hiPSC-mediated neurodevelopmental disease modeling, while hampered by intra- and inter-patient variability (*Brennand et al., 2012*), has capitalized on the medium's unique ability to provide: (1) a model of human development at pathologically-relevant time points, (2) an unlimited source of cells for examination, and (3) tissue-specific cell types which share the genetic background of the somatic cell donor. While initial efforts were focused on monogenic neuropsychiatric disorders (*Marchetto et al., 2010*; *Urbach et al., 2010*; *Paşca et al., 2011*), recent work has expanded to include early investigations of gene-environment interactions (*Hogberg et al., 2013*) and complex polygenic disorders (*Brennand, 2011*; *Shcheglovitov et al., 2013*; *Yoon et al., 2014*; *Mariani et al., 2015*; *Madison et al., 2015*).

hiPSCs have offered a novel window into human-specific alterations in neurodevelopment for a number of disorders. Previous work has shown that neurons derived from 16p11.2 patient hiPSCs exhibit abnormal somatic size and dendritic morphology (*Deshpande et al., 2017*). Interestingly, transcriptional and phenotypic abnormalities have also been observed in hiPSC-derived cortical neural stem cells derived from individuals with idiopathic ASD (*Marchetto et al., 2017*; *Schafer et al., 2019*). The 16p11.2 deletion has widespread effects on signaling pathways that underpin critical neural progenitor functions, including proliferation and fate choice (*Pucilowska et al., 2018*), yet 16p11.2 CNV-related alterations in gene expression patterns have not been examined in early neuroepithelial precursors (radial glia), the stem and progenitor cells of the developing cortex.

Here, we report the derivation, neural differentiation, and transcriptomic characterization of hiPSCs reprogrammed from donors with the 16p11.2 CNV. In total, 65 hiPSC clones were generated from 13 donors. Ten donors harbor a microdeletion at the 16p11.2 chromosomal locus and three donors carry a microduplication. All lines reported here are available through the Simons Foundation Autism Research Initiative (SFARI) (https://sfari.org/resources/autism-models/ips-cells). As a cautionary note, we also report on the relatively high frequency at which hiPSC clones were found to contain randomly integrated reprogramming vectors, in spite of the use of non-integrating episomal reprogramming strategies. In many clones, this led to the inappropriate expression of reprogramming genes in neural progenitor cells (NPCs), which had a more penetrant impact on genome-wide gene expression patterns than the 16p11.2 CNVs. Moreover, we report that only a subset of genes within the 16p11.2 CNV locus are expressed in early NPCs, and 14 of these genes show significant reduction in RNA abundance in clones that carry the 16p11.2 deletion. We also show that these

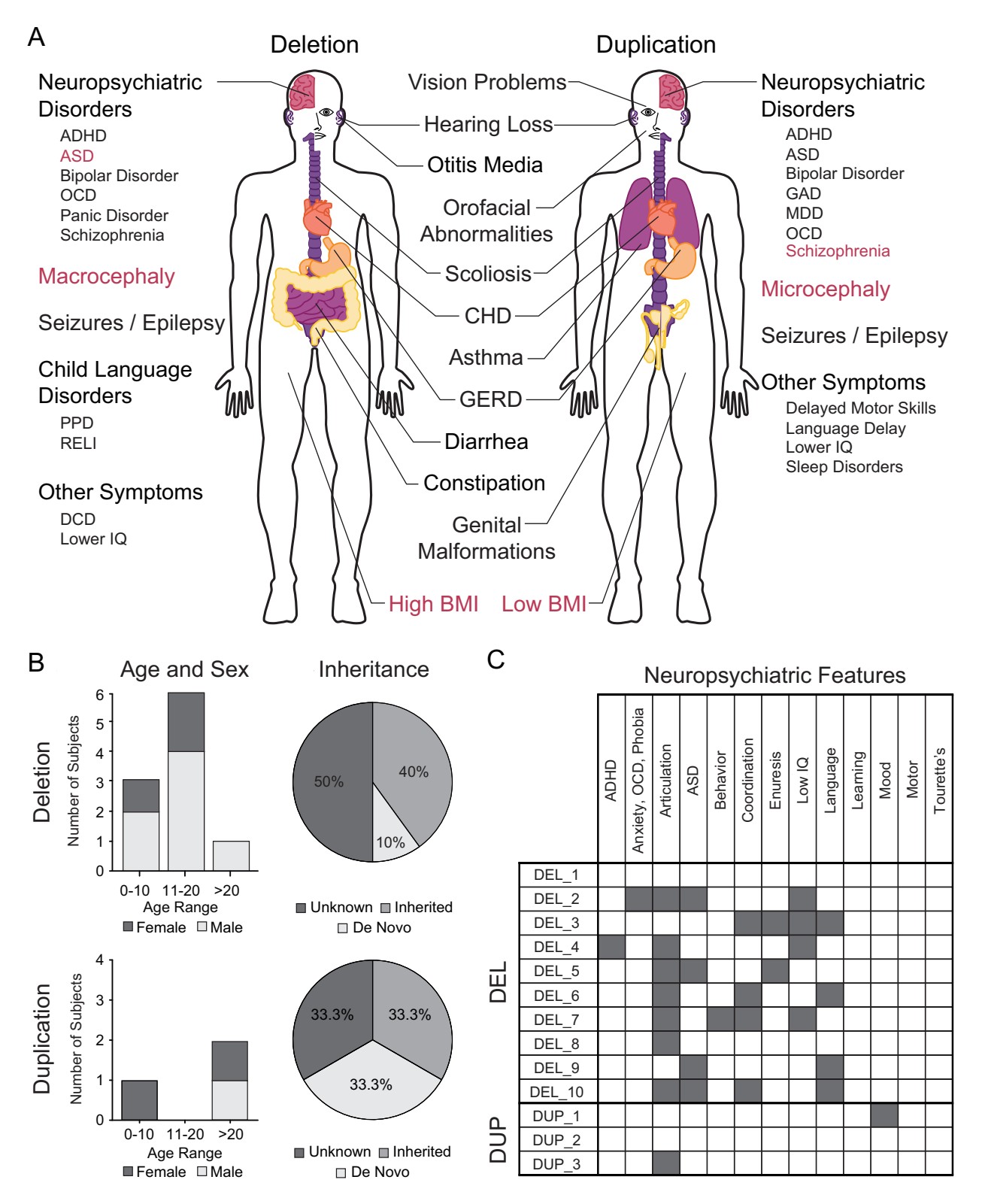

**Figure 1.** Summary of 16p11.2 CNV clinical features and subject demographics. See also *Supplementary file 1*. (**A**) Microdeletions and microduplications of the 16p11.2 chromosomal region are implicated in a collection of aberrant behavioral, physiological, and morphological conditions. Common conditions associated with each copy number variant are listed here. Red text indicates reciprocal phenotypes. Abbreviations: ADHD, attention-deficit/hyperactivity disorder; ASD, autism spectrum disorder; OCD, obsessive-compulsive disorder; PPD, phonological processing

*Figure 1 continued on next page*

*Figure 1 continued*

disorder; RELI, receptive-expressive language impairment; DCD, developmental coordination disorder; CHD, congenital heart disease; GERD, gastroesophageal reflux disease; GAD, generalized anxiety disorder; MDD, major depressive disorder. (B) A summary of age, sex, and mutation inheritance information for individuals with the 16p11.2 CNV whose fibroblasts were reprogrammed into hiPSCs. (C) Neuropsychiatric attributes in fibroblast donors. Additional neuropsychiatric information exists for each individual (see SFARI VIP database, *Supplementary file 1*). Dark gray boxes indicate positive diagnoses, while white boxes represent negative diagnoses.

alterations are accompanied by changes in the expression of 93 additional genes that are not located within the 16p11.2 deletion interval, including genes with known relevance in neurodevelopment. These genes impinge on signaling pathways relevant to recently described phenotypic abnormalities in neurons in vitro (*Deshpande et al., 2017*) and in human carriers in vivo (*Shinawi et al., 2010*; *Qureshi et al., 2014*). Finally, we expand upon these findings by exploring the correlation between gene networks and clinical traits from donors and identify cellular metabolism as a compelling avenue for continued investigation.

## Results

### The 16p11.2 CNV donor population is demographically and phenotypically diverse

hiPSCs were derived from skin fibroblasts isolated from individuals recruited to participate in the Simons Variation in Phenotype (VIP) project. A full spectrum of physiological and neuropsychological evaluations was performed and catalogued throughout their involvement with the project and is available to investigators through SFARI. Here, we present a summary of demographic and diagnostic information for 13 individuals who carry a 16p11.2 CNV and from whom at least two hiPSC clones were generated [*Figure 1B,C*].

Within the cohort of ten deletion donors, there are seven males and three females ranging in age from 6 to 41 years (mean 13.8 ± 9.98 years) [*Figure 1B*]. Seven of the ten donors are probands, one is the father of a proband, and two are siblings of a proband. It is known that four donors carry de novo mutations, and one carries an inherited mutation. Of the three duplication donors, two are female and one is male with ages ranging from 5 to 38 years (23.33 ± 16.80 years). Among the three duplication donors, one is a proband and the other two are a mother (of the aforementioned proband) and a father of a proband from whom hiPSCs were not derived. Of the three donors, one carries a de novo mutation and one carries an inherited mutation.

A battery of cognitive and psychiatric evaluations was performed on each donor. Diagnoses were made in accordance with criteria established in the fourth edition of the Diagnostic and Statistical Manual of Mental Disorders. Among the 16p11.2 deletion donors, the most common diagnosis was articulation disorder (7 of 10 individuals), followed by ASD (4 of 10), an IQ below 70 (4 of 10), language disorder (4 of 10), and coordination disorder (4 of 10) [*Figure 1C*]. None of the 16p11.2 duplication donors were diagnosed with ASD or SCZ, although one individual has articulation disorder while another has mood disorder [*Figure 1C*]. Additional metadata for each donor are included in *Supplementary file 1*.

### 16p11.2 CNV skin fibroblasts were reprogrammed into hiPSCs

Skin fibroblasts from each 16p11.2 CNV donor were transformed with episomal reprogramming vectors that express *SOX2*, *OCT3/4*, *KLF4*, *LIN28*, *L-MYC*, and P53-shRNA (*Okita et al., 2011*; *Figure 2A*). Candidate hiPSC clones were identified by the emergence of tightly packed colonies of cells with high nucleus to cytoplasm ratios and sharp colony margins [*Figure 2—figure supplement 1A,B*]. Immuno-fluorescent staining for four pluripotency markers Nanog, OCT3/4 (also known as POU5F1), TRA-1–60, and TRA-2–49 confirmed that cells in each clone were >95% positive for each marker [*Figure 2—figure supplement 1C–F*]. hiPSC pluripotency was further verified by directed differentiation of the clones into endoderm, mesoderm, and ectoderm lineages, and by assessing changes in pluripotency and lineage marker expression specific to the three germ layers [*Figure 2B*, *Figure 2—figure supplement 1G*]. Additional quality control data for each hiPSC clone are included in *Supplementary file 2*.

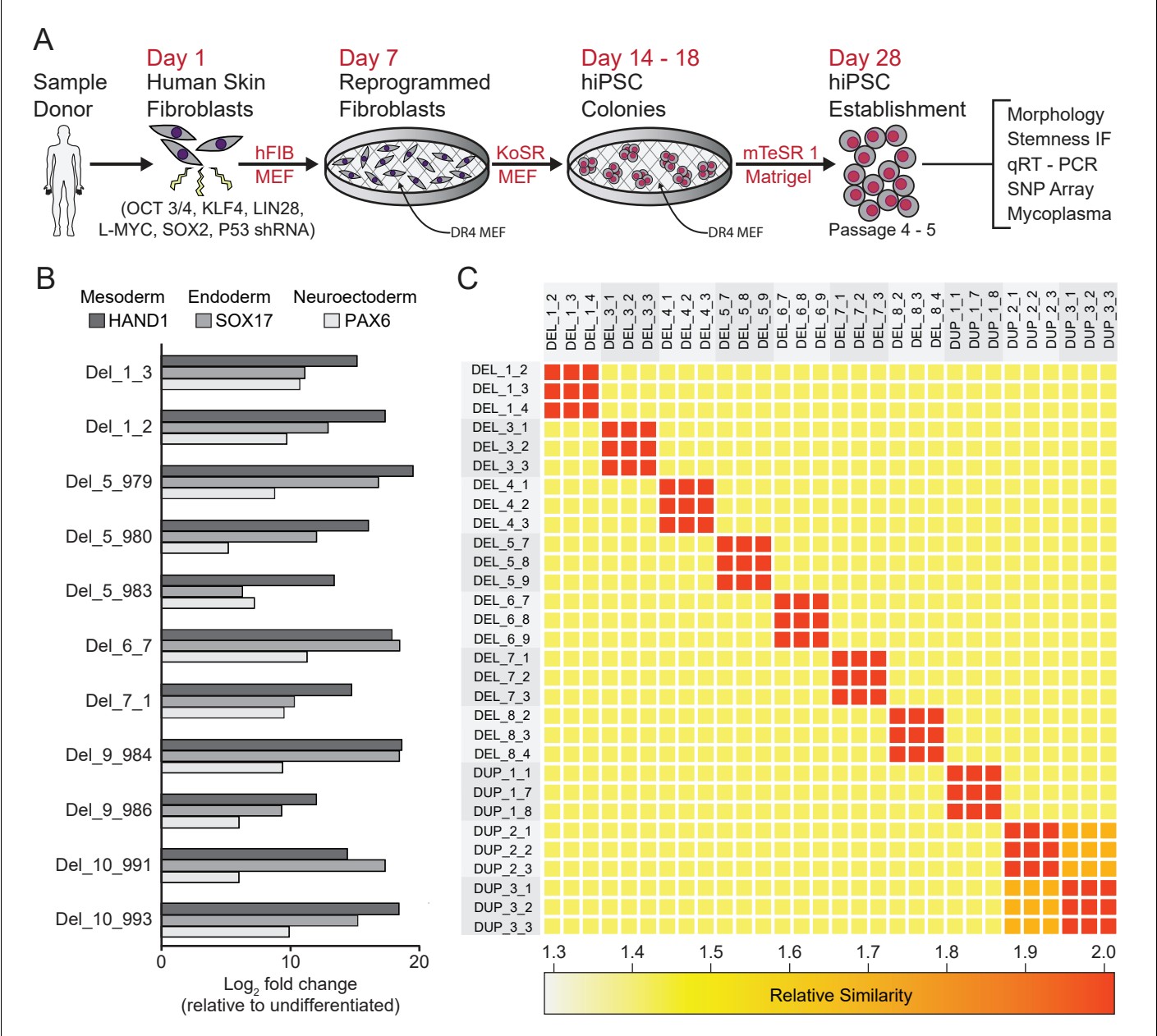

**Figure 2.** Derivation and validation of 16p11.2 CNV hiPSCs. See also *Supplementary files 1*, *2* and *3*; *Figure 2—figure supplement 1* and *Figure 3—figure supplement 1*. (A) Schematic of episomal reprogramming of human fibroblasts into hiPSCs. Abbreviations: hFIB, human fibroblasts; MEF, mouse embryonic fibroblasts; KoSR, KnockOut Serum containing media; IF, immunofluorescence; qRT-PCR, quantitative real-time polymerase chain reaction; SNP, single nucleotide polymorphism. (B) qPCR analysis of lateral mesoderm (Hand1), definitive endoderm (Sox17), and neuroectoderm (PAX6) marker expression following directed differentiation into each respective lineage. (C) SNP-based similarity matrix illustrating the degree of familial relatedness across a subset of hiPSC clones. Increased similarity between clones is indicated in red. Family members share a larger number of SNPs (orange) than unrelated individuals (yellow).

The online version of this article includes the following figure supplement(s) for figure 2:

**Figure supplement 1.** Fibroblast reprogramming and pluripotency validation.

The majority of hiPSC clones also underwent analysis by single nucleotide polymorphism (SNP) array (*Simmons et al., 2011*; *Zhu et al., 2011*) to characterize the degree of relatedness between clones, ploidy, and the presence of additional CNVs. All clones were confirmed to be diploid, and, with the exception of two sets of clones from related donors, all clones were unrelated to each

other, verifying that archived clones are unique to the respective donor and that inadvertent cross-contamination of cells from independent donors did not occur [*Figure 2C*]. Genetic similarity was confirmed for the two familial donors (DUP_2 and DUP_3) who were known to be mother and daughter. SNP analysis also confirmed that the SNPs proximal to the 16p11.2 CNV breakpoints were consistent with previously reported values, ranging from genomic coordinates 29.4 Mb to 32.2 Mb (*Weiss et al., 2008*; *Jacquemont et al., 2011*; *Blumenthal et al., 2014*; *Supplementary file 1*). Finally, the SNP analysis revealed that CNVs outside of the 16p11.2 locus exist in several patients; these CNVs are summarized in [*Supplementary file 3*].

## 16p11.2 CNV hiPSCs form patterned cortical neural rosettes

40 of the 65 clones were subjected to an adaptation of a monolayer dual-SMAD inhibition protocol (*Shi et al., 2012*) that promotes the formation of dorsal forebrain patterned neural rosettes and a variety of neuronal and glial subtypes [*Figure 3A*]. Flow cytometry confirmed the rapid extinction of OCT4 expression and induction of the radial-glial marker paired box 6 (PAX6) within 7 days [*Figure 3—figure supplement 1A*]. Further differentiation of the cells resulted in the formation of neural rosettes composed of radially arranged NPCs, which is considered to be an in vitro recapitulation of both the cellular identity and morphology of radial glia in the developing neural tube (*Wilson and Stice, 2006*; *Elkabetz et al., 2008*; *Figure 3B*). There were no subjective differences between WT and 16p11.2 CNV clones in their ability to form rosettes [*Figure 3—figure supplement 1B*]. After 26 days of differentiation, cells were fixed and stained for a panel of nuclear and cytoplasmic markers to evaluate radial glial cell identities. In time-matched wild-type (WT) control rosettes and 16p11.2 CNV rosettes, PAX6-positive radial glia were similarly arranged in radial clusters [*Figure 3B*, *Figure 3—figure supplement 1B*]. The establishment of normal epithelial polarity was inferred by the strong apical localization of N-Cadherin (NCAD)-positive adherens junctions [*Figure 3B*, *Figure 3—figure supplement 1B*]. The apical end feet of the radial glia were positive for the tight junction marker zonula occludens-1 (ZO-1) as well as the PAR complex protein atypical protein kinase C zeta (aPKCζ) which is consistent with the establishment of radial glial apical-basal polarity. Additionally, mitotic cells, indicated by phospho-histone H3 (pHH3), were found within rosettes proximal to Pericentrin labeled centrosomes which were predominantly located in the center of the rosette, a localization that mimics the apical localization of mitotic cells observed in radial glia in vivo (*Kriegstein and Alvarez-Buylla, 2009*). A subset of clones was further differentiated to assess their ability to generate immature neurons. After 45 days of monolayer differentiation, cells exhibited characteristically long, branching neuron-specific class III β-tubulin (TUJ1) positive projections, and a small portion of young neurons was positive for neuronal nuclear protein (NEUN) [*Figure 3—figure supplement 1C*].

To more thoroughly evaluate the patterning and differentiation of cells within the neural rosettes, total mRNA was collected from hiPSC-derived neural rosettes after 22 days of differentiation and genome wide transcriptomes were evaluated. Our initial analysis prospectively examined markers that should be enriched or depleted in neurectoderm of the dorsal telencephalon (*Rubenstein et al., 1998*). All clones showed strong enrichment for anterior- and dorsal-specific markers, confirming the establishment of anterior cortical identities in culture [*Figure 3C*].

## Reprogramming vector integration induces transcriptional abnormalities

We used principle components analysis (PCA) to evaluate transcriptional variance across the differentiated clones. Surprisingly, PCA showed strong segregation of the clones into two distinct populations that were entirely unrelated to 16p11.2 genotype [*Figure 4A*]. With further analysis, we found that principle components 1 and 2 were associated with elevated expression of OCT3/4, which was subsequently confirmed by quantitative polymerase chain reaction (qPCR) and quantitative real time PCR (qRT-PCR) to be due to the unanticipated random integration, and inappropriate expression, of episomal reprogramming vectors [*Figure 4—figure supplement 1*].

With a novel pseudo-alignment pipeline, we were able to deconvolve counts for a given reprogramming gene to those generated from the genome and those generated from an integrated plasmid [*Figure 4B,C*]. To identify the relative contribution of transcripts from the genome compared to the plasmid integrant, we re-aligned the raw sequencing reads to a composite reference genome

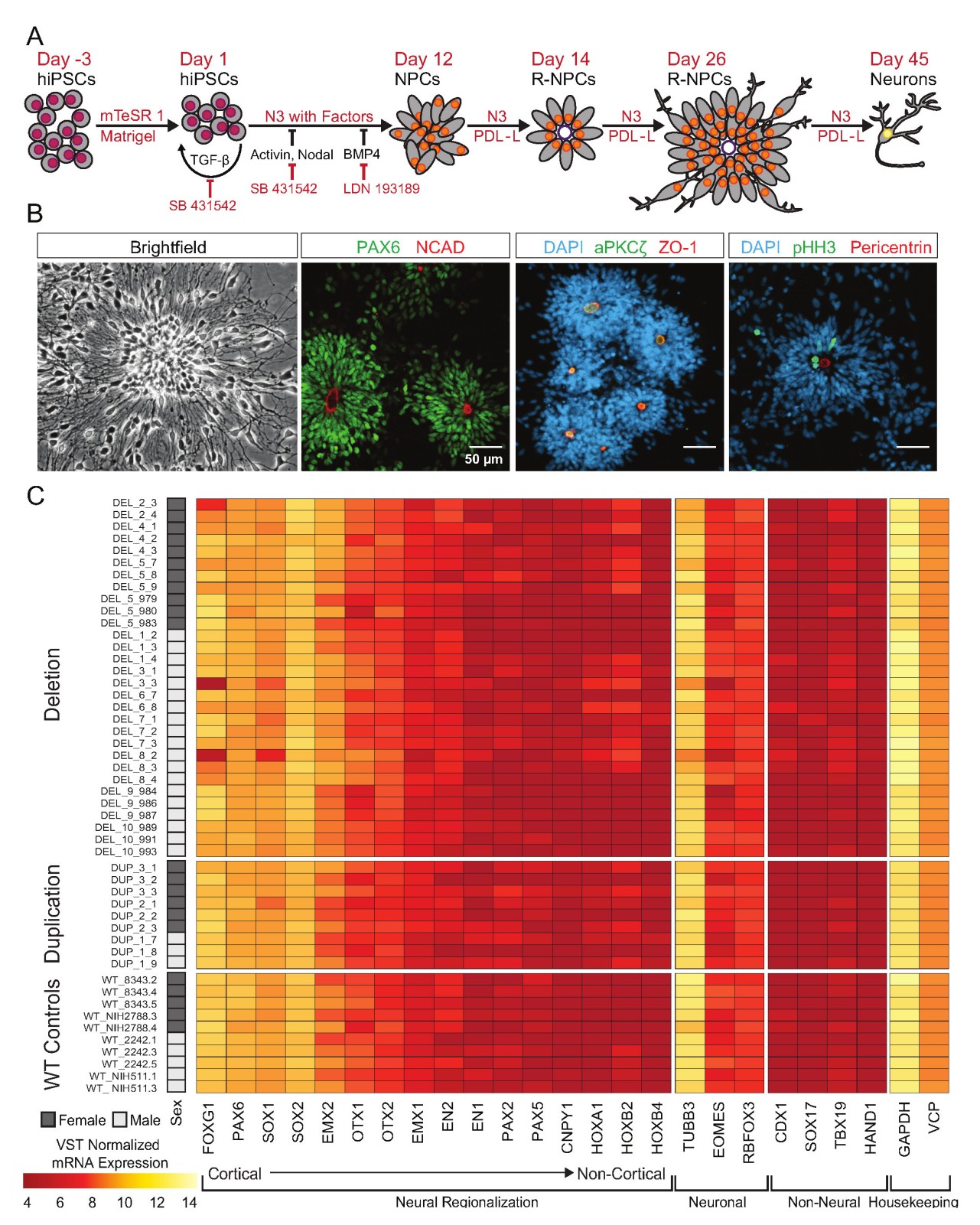

**Figure 3.** hiPSCs differentiate into cortical neural lineages. See also *Supplementary file 2*; *Figure 4—figure supplement 1*. (**A**) Schematic of neural differentiation of hiPSCs into cortical progenitor cells and neurons utilizing dual SMAD inhibition. Abbreviations: N3, basal neural differentiation medium, PDL-L, Poly-D-Lysine and Laminin coating; R-NPCs, radial NPCs. (**B**) Day 26 neural rosettes show the typical radially arrayed clusters of neural progenitor cells in brightfield micrographs. Rosettes are composed of PAX6-positive radial glia encircling a NCAD-positive, ZO-1-positive, and aPKCζ-

*Figure 3 continued on next page*

*Figure 3 continued*

positive apical adherens complex. Cells currently undergoing M-phase of mitosis, indicated by pHH3, are predominantly localized around Pericentrin positive centrosomes at the apical end foot of radial glia (scale bars, 50 μm). Representative rosettes are shown from left to right for (WT_8343.2, WT_8343.2, WT_8343.4, and WT_2242.5). (C) Normalized transcript expression levels of neural regionalization candidate genes generated from RNA-Seq data, ordered from rostral to caudal cell fates, followed by general neuronal and non-neural cell fates, and housekeeping genes. Sex and Genotype status are indicated on the left.

The online version of this article includes the following figure supplement(s) for figure 3:

**Figure supplement 1.** hiPSCs differentiate into NPCs and neurons.

consisting of a human reference genome (Ensembl GRCh38.93) and plasmid sequences inserted as extra chromosomes [*Figure 4B*]. With this method, we were able to identify whether the integrated plasmids were transcriptionally active, as well as pinpoint potential transcriptional effects of the integration. Notably, the integration-free hiPSC-derived NPCs express *LIN28a*, *MYCL*, and *SOX2*. Although several clones had integrated plasmids that encode each transcript, the reprogramming vectors did not significantly contribute to the abundance of the respective mRNAs [*Figure 4C*]. Conversely, approximately 10% of *KLF4* transcripts detected were expressed from the plasmid and 40% of *POU5F1* (*OCT3/4*) transcripts were plasmid-derived. Although all tested hiPSC clones were competent to differentiate into cortical neural rosettes, we hypothesized that the presence of *POU5F1* integrant may affect the self-renewal capacity of the differentiated NPCs (*Niwa et al., 2000*; *Boyer et al., 2005*; *Wang et al., 2012*). Notably, We observed a significant increase (p=0.0385) in the proportion of pHH3-positive cells in the DEL Int+ differentiated NPCs affected by random integration of the *POU5F1* plasmid [*Figure 4—figure supplement 2A*].

To more clearly define the impact of unanticipated episomal vector integration and to determine if clones harboring integrated reprogramming vectors could be used for further analysis, a full RNA-seq analysis was performed on all clones. Differential expression analysis was used to identify alterations due to integration. After correcting for the effects of sex, 16p11.2 genotype, and sequencing batch [*Figure 4—figure supplement 2B–E*], we identified 3612 differentially expressed (DE) genes with an adjusted p-value<0.05, of which 1739 genes (48.15%) were downregulated in integration-containing (Int+) clones relative to integration-free (Int-) clones. Transcripts attributed via alignment to integrated plasmids were detected and significantly elevated for all three reprogramming plasmids, as well as in transcripts associated with genomic *POU5F1*, *KLF4*, *SOX2*, and *LIN28A*. There was no statistically significant difference in *MYCL*. All these DE transcripts were increased in the INT + clones, except for the two non-POU5F1 bearing plasmids, which were detected as significantly downregulated in INT+ clones. Given the low baseline expression of these two plasmids relative to the POU5F1-bearing plasmid or the genome-associated transcripts, this DE signature may be dominated by noise generated during the alignment procedure.

The full list of DE genes impinges on major cell functions, such as numerous HOX genes (e.g. *HOXA2*, *HOXB2*), regulators of cell morphology (*PCDH8*, *ACTN3*), cell cycle (*CDKN1A*, *AKT3*, *STAT3)*, and major signaling pathway effectors (e.g. *TP53*, *WNT1*). Within the top 100 DE genes [*Figure 4D*], a clear transcriptional difference emerges that separates Int+ from Int- clones, regardless of 16p11.2 CNV. Although less acute, differences in cell fate acquisition are also observed when clones are sub-divided into Int+ and Int- [*Figure 4—figure supplement 2F*]. To computationally explore the cellular functions that might be implicated erroneously by these transcriptional changes, we applied Gene Set Enrichment Analysis (GSEA) to further probe biologically significant sets of genes that are enriched at the extremes of a ranked list of p-values. At a false detection rate (FDR) < 0.25, over 52 functions were enriched among low p-value/upregulated genes including pathways related to embryonic development and regionalization, HOX gene activation, transcription, and translation [*Figure 4E*]. 355 functions were enriched among low p-value/downregulated genes including pathways related to cell death, cell adhesion, and the JAK/STAT cascade [*Supplementary file 4*]. Based on these results, we concluded that the transcriptional effects of integration would confound experimentally relevant phenotypes caused by 16p11.2 CNVs, and the Int+ clones were excluded from subsequent analyses. Importantly, an insufficient number of Int- 16p11.2 duplication clones remained (n = 4) for meaningful analysis, so further studies focused exclusively on 16p11.2 deletion clones.

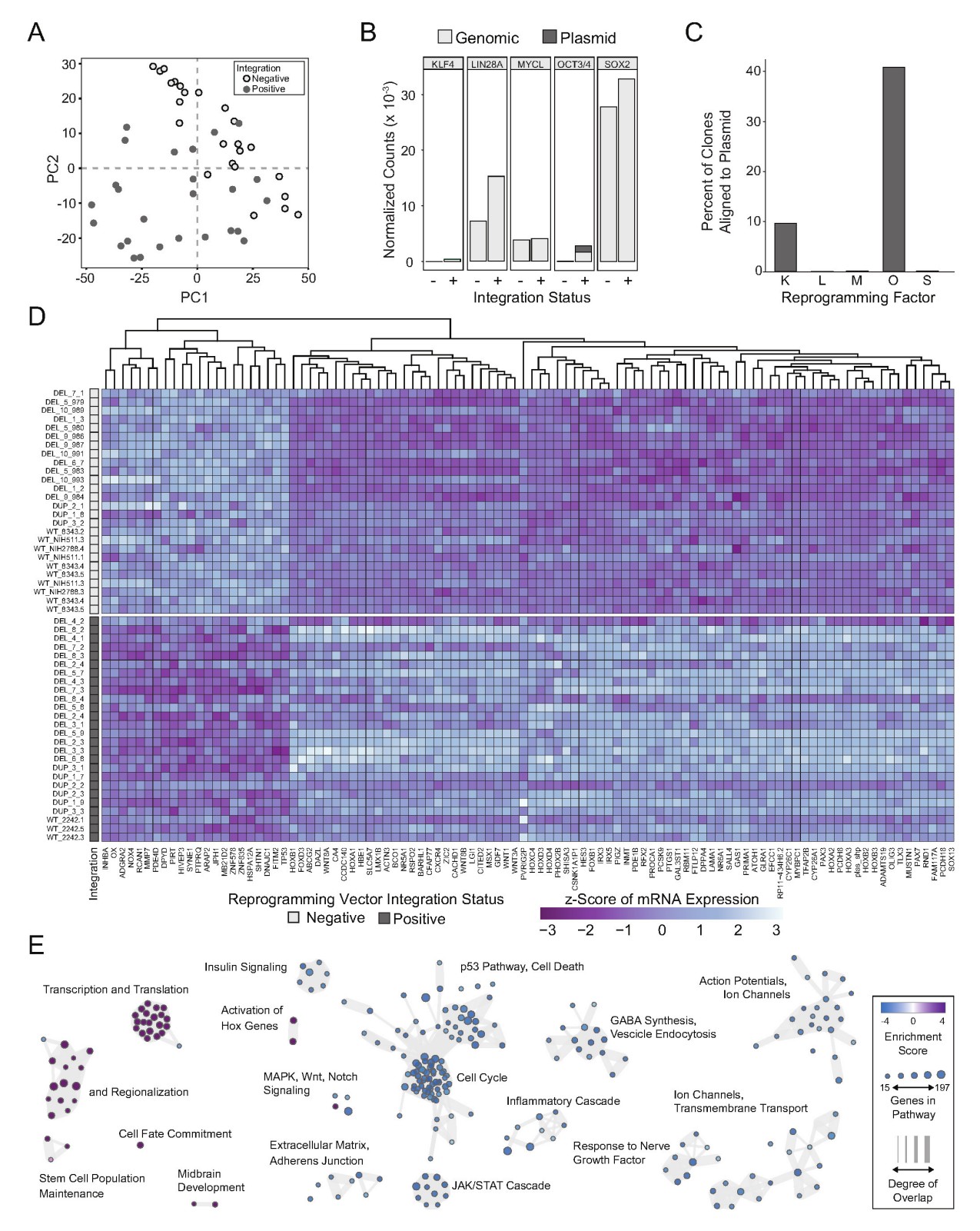

**Figure 4.** Integration and expression of reprogramming vectors generates pronounced artifacts in the transcriptome. See also *Supplementary files 2 and 4*; *Figure 4—figure supplements 1* and *2*. (A) PCA of variance-stabilized count data before batch correction reveals that samples cluster by integration status within the first two PCs. Axes represent the first two principal components (PC1, PC2). (B) Reprogramming factor expression from reads pseudo aligned to the human genome or to plasmid sequences in Int- and Int+ clones. Y-axis represents estimated counts normalized by size

*Figure 4 continued on next page*

Figure 4 continued

factor. The absence of plasmid-aligned transcripts for most genes is indicated by the absence of dark gray segments for each bar (with the exception of *OCT3/4*). (C) Percentage of total *KLF4* (K), *LIN28A* (L), *MYCL* (M), *OCT3/4*(O), or *SOX2* (S) counts pseudo-aligned to plasmid in Int+ clones. Y-axis represents the percentage of counts reported in (B). (D) Heatmap of gene expression represented as Z-scores for the top 100 differentially expressed genes in Int- and Int+ clones as identified with DESeq2. Counts were normalized and scaled using a variance-stabilizing transformation (VST) implemented by DESeq, with batch effect correction using limma. Integration status is visualized on the left (integration free clones on top, light gray indicator). (E) GSEA analysis of DESeq output identified biological functions potentially impacted by cryptic reprogramming vector integration. Individual nodes represent gene lists united by a functional annotation; node size corresponds to the number of genes in pathway, and color reflects whether the pathway is upregulated (purple) or downregulated (blue). Only nodes with significant enrichment in our DESeq output are displayed. The number of genes shared between nodes are indicated by the thickness of their connecting lines. For ease of visualization, individual node labels have been replaced with summary labels for each cluster.

The online version of this article includes the following figure supplement(s) for figure 4:

**Figure supplement 1.** Presence of 16p11.2 reprogramming vector integration and transcripts.
**Figure supplement 2.** Influence of OCT3/4 integration on cell phenotype and transcriptome.

## The 16p11.2 microdeletion affects the transcriptome of hiPSC-derived cortical neural rosettes

RNA-seq data from Int- 16p11.2 deletion (DEL, n = 13) and wild type (WT, n = 7) were re-normalized and analyzed for the potential effects of 16p11.2 deletion in cortical neural rosettes. In addition to explicitly correcting batch effects of sex and clone preparation date, surrogate variable analysis (SVA) was performed to account for both potential patient-of-origin batch effects and additional uncharacterized sources of undesirable variance. PCA revealed that the DEL and WT populations clustered according to genotype within the first two principal components after accounting for batch effect correction [*Figure 5A*]. The known deletion intervals for each of the DEL clones are illustrated in *Figure 5B–D*. There are 56 canonical gene IDslocated in the 16p11.2 locus between positions 28,800,000 and 30,400,000 [*Figure 5D*]. Of these, 14 of the 16p11.2 region genes were differentially expressed, and all were downregulated which is consistent with copy number-dependent effects on mRNA abundance [*Figure 5B*].

In total, 107 DE genes were identified in the DEL samples relative to WT (adjusted p-value<0.05). A full annotated characterization of these genes is provided in the supplement [*Supplementary file 5*]. The most affected 49 DE genes (1.5-fold or greater change) include all 14 DE genes from the 16p11.2 deletion interval [*Figure 5E*]. A scatterplot of normalized relative mRNA abundance for all transcripts in WT and DEL clones shows relatively tight correlations across genotypes with the deletion interval genes showing a more pronounced change relative to DE genes from other loci in the genome [*Figure 5F*]. Using qPCR we confirmed that a subset (*KCTD13, TAOK2, MAPK3* and *SEZ6L2*) of 16p11.2 region genes that were found to be DE from our transcriptome analysis were indeed downregulated in each of the DEL samples relative to WT [*Figure 5—figure supplement 1*]. In addition, the individual DEL clones showed strong concordance with the direction of change and approximate amplitude of change for each DE gene within each clone [*Figure 5—figure supplement 2*].

The entire collection of DE genes [*Figure 5G*] are implicated in a variety of functions. DAVID gene enrichment analysis shows that several disease functions related to neuropsychiatric disorders are associated with more than six genes in the differential expression list [*Figure 5—figure supplement 3A*]. Additionally, the differentially expressed gene list is statistically significantly enriched for genes associated with ASD, including two genes outside the 16p11.2 region (LHFPL3, CNTNAP2) [*Supplementary file 6*]. Although the enrichment does not reach significance following multiple hypothesis testing comparison, it is worth noting that additional genes are implicated in psychiatric disorders (GABRE, GNRH1, FMO1, FLG, SELE, NQO2). Genes relevant to neuroepithelial development are also identified, such as the Erk1/2 MAPK signaling pathway (MAPK3, RAF1, SEMA4G), synapse growth and regulation (CNTNAP2, CALML6, C1QL3, GABRE, PRRT2, SEZ6L2, DOC2A), cytoskeletal organization and cell adhesion (SELE, TEKT4, TUBGCP4, KLHL4), cell cycle (PAGR1, PIWIL3), and fate choice (GSC, OTP, PAX3) [*Figure 5—figure supplement 3B*]. Although the basic mechanisms that mediate neuroectodermal specification and formation of neural rosettes formation was not measurably altered, the DE genes within these early stem and progenitor cells suggest that

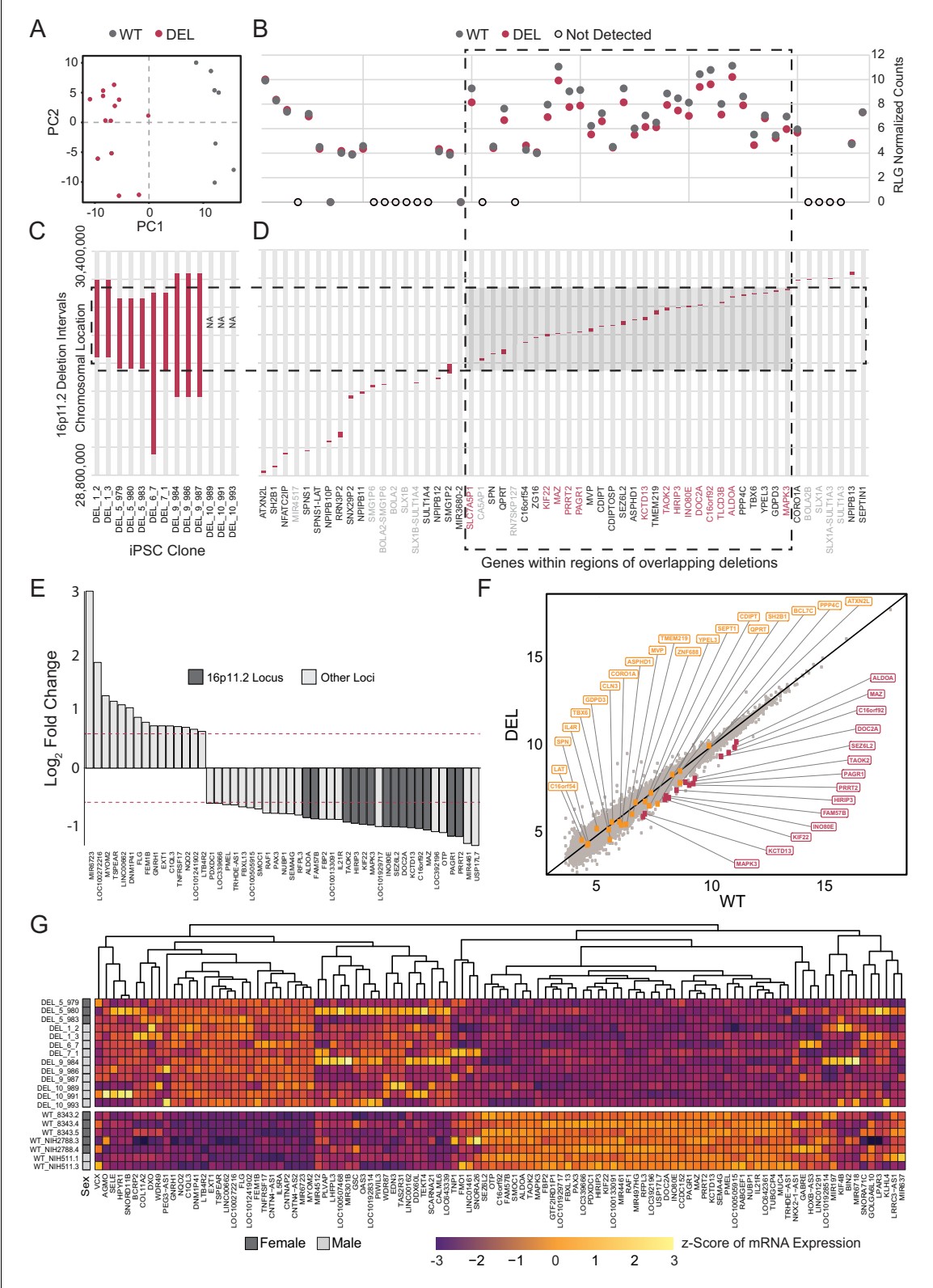

**Figure 5.** Deletion intervals and differential expression of genes at the 16p11.2 locus. See also *Supplementary files 5* and *6*; *Figure 5—figure supplements 1*, *2*, *3* and *4*. (**A**) PCA of variance-stabilized count data after normalization and batch correction reveals that samples cluster by 16p11.2 deletion status within the first two PCs. Axes represent the first two principal components (PC1, PC2). (**B**) RLG normalized counts of each transcript within the 16p11.2 interval. WT = black symbols, DEL = red symbols, transcripts not detected = gray symbols. (**C**) Known deletion intervals in

*Figure 5 continued on next page*

*Figure 5 continued*

integration-free hiPSC clones that were included in the RNA-seq analysis of differentially expressed genes within the 16p11.2 locus. NA, breakpoint information was not available from the Simons Foundation. (D) Canonical gene symbols located between chromosome 16 location 28,800,000 and 30,400,000. Transcripts that reach significance as differentially expressed between WT and DEL clones (FDR < 0.05) are indicated in red. Labels for transcripts that were below detection limits are marked in light gray. (E) Differentially expressed genes that are up- or downregulated at least 1.5-fold. Red lines represent threshold of 1.5-fold change. Genes falling within the 16p11.2 deletion region are highlighted. (F) VST-normalized and batch corrected expression for all genes across all WT clones (X-axis) and DEL clones (Y-axis). Highlighted points represent 16p11.2 region genes that were either called differentially expressed (Red) or not differentially expressed in our pipeline (Orange). (G) Heatmap of gene expression for all the differentially expressed genes identified with DESeq2. Fill values represent counts that have been normalized and scaled using a variance-stabilizing transformation implemented by DESeq, and batch effect corrected using limma and SVA. Sex of the subject is indicated on the left.

The online version of this article includes the following figure supplement(s) for figure 5:

**Figure supplement 1.** Validation of differentially expressed 16p11.2 interval genes.
**Figure supplement 2.** Concordance of differentially expressed gene changes across individual clones.
**Figure supplement 3.** DAVID gene enrichment analysis for differentially expressed genes.
**Figure supplement 4.** Differential expression analysis with a linear mixed model to account for shared patient identity across clones.

16p11.2 deletion may impact many functions important to neurodevelopment and neuropsychiatric disease.

We further validated these results by identifying differentially expressed genes using an orthogonal approach to compare transcriptomic features for clones from each donor. Using a linear mixed model implemented by the DuplicateCorrelations() function of limma/voom to account for shared patient identities across clones, we identified 40 genes as differentially expressed, 17 of which were in common with the genes presented in the first differential expression analysis [*Figure 5—figure supplement 4A*]. The interpretation of this gene list was consistent with our first analysis. Ten 16p11.2 region genes were statistically significantly downregulated in DEL relative to WT. Alterations in the expression of several miRNAs (e.g. miR-6723) were still apparent. The differentially expressed gene list retained genes critical for synaptic development (e.g. *GABBR1, TAOK2, DOC2A),* cytoskeletal organization (e.g. *PDXDC1),* metabolism (e.g. *ACSM4),* and the cell cycle (e.g. *PAGR1, NUBP1),* and the Erk1/2 MAPK pathway (e.g. *RAF1, RGL2)* [*Figure 5—figure supplement 4B*].

Next, we investigated whether transcriptional patterns existed within DEL clones that correlate with any of the noted clinical features in DEL donors. Weighted Gene Co-Expression Network Analysis (WGCNA) was used to identify patterns of gene expression that significantly correlated with clinical features for each donor of the DEL clones. In this analysis, we restricted our module identification to only Int- DEL clones for which clinical data was available. 123 modules of gene expression were identified in the data, of which 20 modules had a significant association at least one variable analyzed [*Figure 6A*]. Of these, 15 showed significant association with a documented clinical phenotype and four modules had highly significant correlation to patient Age, Height Z-score, Weight Z-score, and Head Circumference Z-score.

We next asked what functional role the genes in these modules might have by performing GSEA. The genes identified in each WGCNA module are unsigned and therefore each module contained genes whose expression may be positively or negatively correlated with traits.We chose to rank genes by magnitude of fold change in the top half of Weight_Z scores relative to low Weight_Z. This ranked list was submitted to GSEA, which identified 69 gene sets as upregulated in individuals in the top half of BMI scores [*Supplementary file 7*]. Of these, the majority of enriched genes fell within metabolic processes (e.g. *ADRA1D, SMARCA1, TRIAP1)* [*Figure 6B*]. Notably, a significant fraction of enriched pathway categories also included signaling pathways and the regulation of gene expression (e.g. *FGF1, SOX11, BMP10, KLF7, HOXA1, HOXB7).* Although metabolic genes represent the majority of significantly enriched pathways in this GSEA, the highest normalization scores were observed in general cellular processes and signaling pathways [*Figure 6C and D*].

Of the four modules related to body mass and language, the pink4 module had both high significance and high correlation with key DEL clinical phenotypes [*Figure 6E* and *Figure 6—figure supplements 1, 2*]. Notably, pink4 was highly positively correlated with donor Age and Weight Z-score, and negatively correlated with Height Z-score, and Head Circumference Z-Score. Interestingly, it was one of the few modules significantly correlated with the Language trait and exhibits a very high

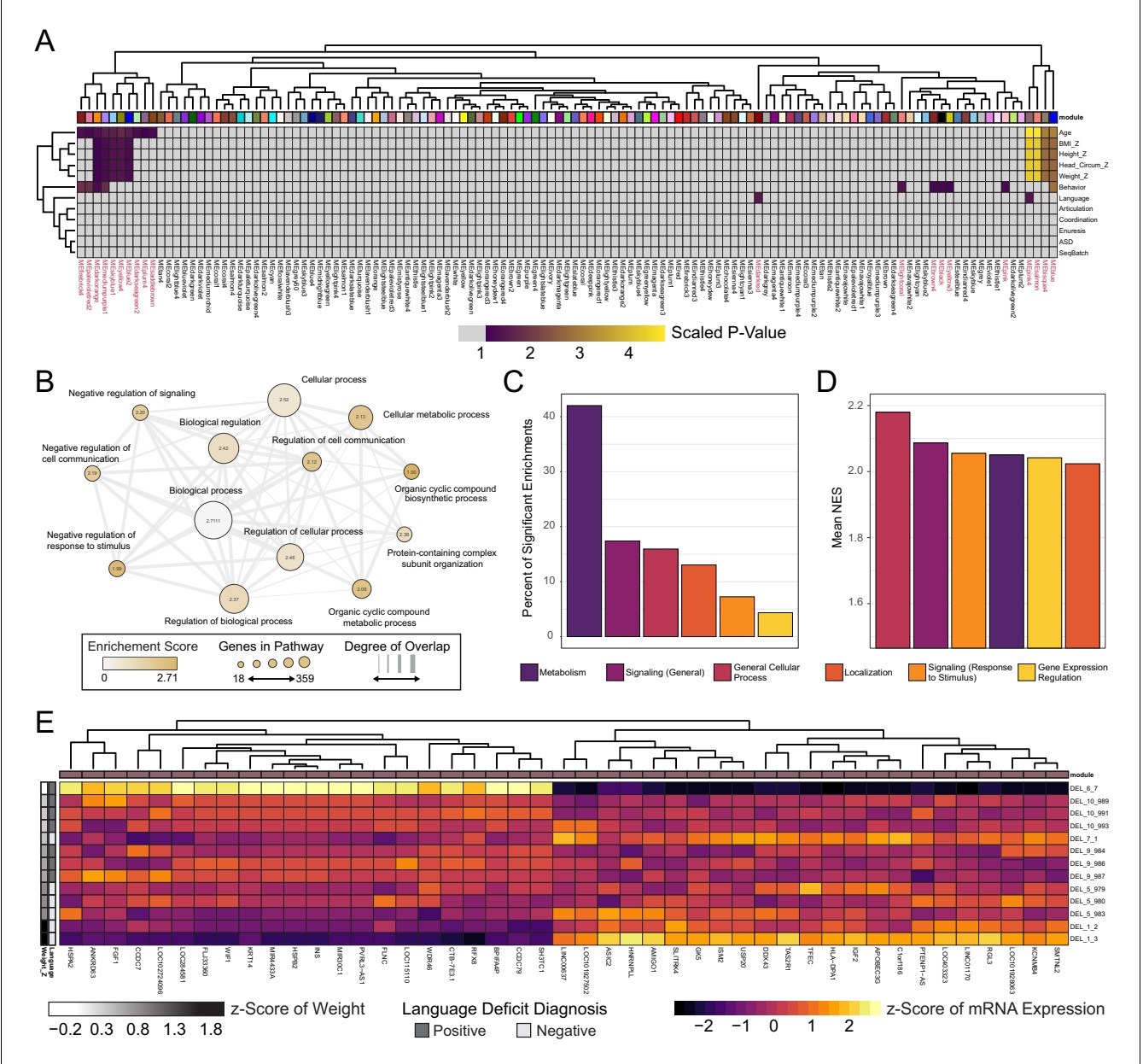

**Figure 6.** WGCNA reveals modules of co-expressed genes in integration-free clones that correlate with patient clinical features. See also *Supplementary file 7*; *Figure 6—figure supplements 1*, *2* and *3*. (A) Heatmap of p-values assessing the significance of module-trait correlations. Values represent a scaled p-value equal to (−1 * log10(p-value)). P-values that fall outside of the significance threshold of p<0.05 are colored gray. WGCNA-produced module color labels are annotated on the X-axis, with red text indicating 20 modules with p<0.05. (B) Depiction of annotations identified as statistically significant (FDR < 0.25) in GSEA for the set of genes identified by WGCNA as the gene networks within the clinical trait-associated modules with highest significance: pink4, salmon, bisque4, and blue (modules represented in the last four columns of panel A). (C) Categories of pathways identified as upregulated among significantly trait-associated module genes by GSEA according to frequency. Enriched pathways identified by GSEA were assigned to categories based on their Gene Ontology relations. (D) Categories of pathways identified as upregulated among significantly trait-associated module genes by GSEA according to normalized enrichment score (NES). Enriched pathways identified by GSEA were assigned to categories based on their Gene Ontology relations. (E) Heatmap of scaled VST-normalized, batch-corrected expression values for genes identified as members of the pink4 module by WGCNA. Phenotype annotations are indicated on the Y-axis.

The online version of this article includes the following figure supplement(s) for figure 6:

**Figure supplement 1.** Visualization of module membership (MM) and gene-trait significance (GS) for modules with statistical significance.

**Figure supplement 2.** Individual sample outliers drive phenotype correlation in three significant modules.

**Figure supplement 3.** Module correlation with donor clinical information.

magnitude of correlation. The high correlation between gene module membership and significance to not only Weight_Z but also Language may unexpectedly reflect alterations in a common gene module that influence clinical features of 16p11.2 biology which have not previously been linked [*Figure 6—figure supplement 3*]. Of the 45 genes with high membership scores for the pink4 module [*Figure 6A and E*], several functions are potentially impacted, including components of the MAPK (*FGF1*) and Wnt (e.g. *WIF1*) signaling pathways, as well as mediators of cell growth and differentiation (e.g. *AMIGO1*).

## Discussion

Current investigations into the etiology of neurodevelopmental disorders have been limited by an inability to recapitulate the developmental trajectory of the human brain under controlled laboratory conditions. hiPSCs offer a unique opportunity to study these early developmental mechanisms. Here, we report the generation and banking of publicly available resource of 65 hiPSC lines with CNVs at the 16p11.2 chromosomal locus. Furthermore, we employ this resource to identify transcriptional alterations that accompany deletion of the 16p11.2 locus in hiPSC-derived NPCs. 16p11.2 CNVs remain one of the most commonly identified variants associated with ASD and, in addition to the previously described links with ASD and SCZ, 16p11.2 CNVs are implicated in intellectual disability, language disorders, ADHD, motor disorders, and epilepsy (*Shinawi et al., 2010*; *Rosenfeld et al., 2010*; *Hanson et al., 2015*).

The hiPSC lines examined here were generated from biospecimens in the Simons Foundation VIP Collection and represent lines from 10 deletion donors and three duplication donors. These 65 lines are a significant addition to the 14 hiPSC clones recently described by *Deshpande et al., 2017* and include donors that are age and sex matched across genotypes. 16p11.2 CNVs are implicated in a spectrum of non-CNS phenotypes, including those of the heart, kidneys, digestive tract, genitals, and bones (*Zufferey et al., 2012*). The recruited individuals for this study encompass a broad spectrum of physiological attributes and psychiatric diagnoses. A substantial number of donors is required to verify gene-specific phenotypes in hiPSC models due to intra- and inter-individual variability between clones (*Brennand et al., 2012*). Clone-to-clone variability, in part due to incomplete epigenetic remodeling and the resultant influence of epigenetic memory (*Polo et al., 2010*; *Kim et al., 2010*; *Lister et al., 2011*), may confound conclusions built upon line-to-line comparisons (*Onder and Daley, 2012*). Additionally, generating large genomic rearrangements in isogenic control lines through genetic engineering strategies such as the CRISPR/Cas9 system is technically challenging (*Grobarczyk et al., 2015*; *Tai et al., 2016*). The availability of a large number of clones provides an opportunity to explore phenotypes that may be relevant to both CNV-specific effects and the influence of an individual's genetic background on the penetrance of a given phenotype. Importantly, this hiPSC collection contains at least three independently derived clones for most donors. It follows that this resource may significantly improve the field's ability to identify morphological, physiological, and circuit-level characteristics at different stages of development that contribute to the etiology of 16p11.2-related phenotypes.

The large number of genes within the 16p11.2 region, and their wide-ranging functions, make it challenging to pinpoint specific genes relevant to neuropsychiatric disease. One of the goals of this work was to determine which 16p11.2 genes are expressed in early cortical development, as well as which show altered expression following the deletion of one allele. Clones that were free of integrated reprogramming vectors represented genomes from six individuals carrying 16p11.2 deletions. Of these, the deletion intervals were known for five subjects [*Figure 5A*]. We found that all 14 DE genes within the canonical 16p11.2 interval were located within a shorter interval of the chromosome where deletions overlapped for the five subjects with known breakpoints. This clustering to a sub-interval of the locus most likely reflects the fact that the majority of clones analyzed were missing one allele of each DE gene in this region. It is likely that clones with larger deletions may also show copy-number-dependent expression for genes outside of the shared interval. For example, it is reasonable to suspect that genes unique to the much larger deletion present in DEL_6_7 are downregulated, but that the difference is diluted in the current analysis by clones with normal copy number in this region.

The data presented can be used to confidently identify genes within the 16p11.2 locus that are expressed in early NPCs. Of 56 genes within and flanking the deletion intervals of the clones

evaluated, 15 are expressed at or below the limits of detection. Although these genes may play roles at later stages of brain development, they seem unlikely to be relevant for the early patterning and initial establishment of neuronal subtype identity. Conversely, it also stands to reason that the remaining 41 genes are likely to have roles in early cortical development that may be affected when copy number is altered. Of these, 14 are differentially expressed and all are downregulated in DEL: *MAZ, DOC2A, PRRT2, PAGR1, TAOK2, KIF22, HIRIP3, ALDOA, C16orf92, INO80E, MAPK3, KCTD13, SEZ6L2,* and *FAM57B.* Their statistically significant downregulation suggests that these 16p11.2 loci are worthy of increased scrutiny as genes that may commonly influence developmental outcomes across individuals that share this deletion interval. Of particular note is the 16p11.2 region gene *MAPK3,* as its protein product Erk1 is known to interact with several other observed DE genes (e.g. *SELE, CALML6, RAF1, SEMA4A).* This finding supports existing work suggesting that the Erk1 MAPK pathway may drive phenotypic abnormalities associated with the 16p11.2 CNV (*Pucilowska et al., 2018*; *Pucilowska et al., 2015*). Although we detected a decrease in *MAPK3* gene expression in all the 16p11.2 deletion lines [*Figure 5—figure supplement 1A*], we did not observe any changes in NPC proliferation between wild type and deletion lines indicating that *MAPK3* may not influence NPC proliferation in these 16p11.2 patient lines [*Figure 4—figure supplement 2A*]. This is consistent with prior work that also found no significant difference in the proliferation of DEL cells at the NPC stage (*Deshpande et al., 2017*).

In addition to the 14 DE genes within the 16p11.2 locus, there are 93 additional genes that are differentially expressed. Fifteen of these genes are identified as being generally related to psychiatric disease, and nine are more specifically identified as related to ASD [*Figure 5—figure supplement 3A*]. All these genes are listed in the SFARI Gene Scoring Module database (*Abrahams et al., 2013*) and are associated with at least five published reports supporting their role as a candidate gene in ASD. Of these genes, *CNTNAP2,* a neurexin family member that is important excitatory synapse formation and function, is one of the strongest known candidate-genes for ASD risk (*Abrahams et al., 2013*). Several additional DE genes are also relevant to excitatory synapses in mature neurons (e.g. *PRRT2, CALML6, C1QL3, DOC2A, GABRE, C1QL3, CNTNAP2),* and may reflect early perturbations in gene networks relevant to cortical excitatory networks. Previous work reports changes in neurite branching and synapse development in more mature neurons (*Deshpande et al., 2017*), and here, we find genes that may be related to these later changes are differentially expressed well before such neurons emerge in culture.

Future studies might utilize these early transcriptional features to test whether related functions are affected in mature neurons. For example, the 16p11.2 region gene *KCTD13* is important for the maintenance of synaptic transmission in the CA1 region of the hippocampus, with its loss resulting in decreased fEPSP slope and the frequency of mEPSP currents (*Escamilla et al., 2017*). Interestingly, the 16p11.2 region gene *TAOK2* has been shown to influence the differentiation, basal dendrite formation, and axonal projection of pyramidal neurons in the cortex with its deletion resulting in abnormal dendrite formation and impaired axonal elongation in mouse brains (*de Anda et al., 2012*). We found the expression of *KCTD13* and *TAOK2* to be reduced across all deletion lines suggesting that reduced *KCTD13* and *TAOK2* expression in cortical progenitors may impact cortical network development and the maintenance of appropriate synaptic tone in the cortex of autistic individuals.

Other features that the DE gene list may predict include alterations related to movement or locomotion of the cell [*Figure 5G*]. Several DE genes modulate actin and cytoskeletal organization, which could contribute to abnormal cell morphology (e.g. *SELE, TEKT4, TUBGCP4, KLHL4).* Additionally, three downregulated genes are related to glucose metabolism and glycogen synthesis (e.g. *TLCD3B, FBP2, ALDOA),* suggesting that changes in cellular metabolism and energy management may influence outcomes. Finally, we observed that the transcript with the highest fold-change and smallest p-value was aligned the microRNA (miRNA) miR-6723. Although our sequencing protocol was not designed to accurately profile mature miRNAs, the library preparation would capture polyadenylated pri-miRNAs that could later be converted into mature miRNAs (*Cai et al., 2004*). The functions of many of these miRNAs are not well characterized but existing literature supports further investigation. miR-6723 itself is enriched in neurons relative to microglia, astrocytes, and endothelial cells (*Spaethling et al., 2017*). Interestingly, it is also downregulated in term placentas of women with chronic Chagas disease, suggesting a role in cellular proliferation or immune response (*Winn et al., 2007*; *Juiz et al., 2018*). Four other miRNAs appear in the differential expression gene list and have roles in proliferation that may impact progenitor function. Of note, miR-301B impairs

cellular senescence and has a role in the spread of certain cancers (*Ramalho-Carvalho et al., 2017*; *Danesh et al., 2018*). miR-637 inhibits tumorigenesis, and its downregulation may enhance cell proliferation (*Zhang et al., 2011*; *Zhang et al., 2018*).

It is also worth noting that SNP array data show that 4 of the 13 DEL Int- clones contained additional CNVs, including microdeletion in 14q11.1, a microduplication in 14q11.2, and a microdeletion in 7p11.2. All of these CNVs are potentially associated with neurodevelopmental abnormalities (*Zahir et al., 2007*; *Varvagiannis et al., 2014*), and it is possible that these additional CNVs contribute to the transcriptional alterations. However, there is no significant difference in the average fold change of DE gene expression in DEL relative to WT between the subset of clones carrying these additional CNVs and the remaining DEL clones [*Figure 5—figure supplement 1B*]. Although it is important to document all background CNVs for future reference, these CNVs are unlikely to account for the transcriptional changes in the NPCs reported here.

This work also suggests that patient-specific transcriptional abnormalities, in addition to a core 16p11.2 transcript deficiency, may interact to generate a heterogeneous patient phenotype. Our WGCNA analysis identified several modules correlated with phenotypes within 16p11.2 deletion patients [*Figure 6A*]. Interestingly, these signals were related not only to higher order neurological functions, but to physical attributes and body plan. These modules do not contain genes identified as distinguishing DEL from WT samples. 16p11.2 region genes were most frequently observed in the grey module (n = 11 genes), followed by the blue (n = 3) and turquoise (n = 1) module. The grey module also contained the most DE genes (n = 32) in the DE pipeline presented in *Figure 5* and genes identified in the linear mixed model pipeline (n = 16) [*Figure 5—figure supplement 4*]. This is to be expected given that the contrast examined with WGCNA was within DEL patients, rather than comparing all DEL clones with WT clones as in the DE analysis of *Figure 5*. However, it is possible that the gene programs accounting for phenotype heterogeneity within DEL are due to transcriptional programs that exist independent of the 16p11.2 deletion region. This further reinforces the possibility that patient-specific etiology may be due to an interaction of a generalized existing liability posed by the presence of the 16p11.2 deletion interacting with patient-specific features.

Within the four modules most highly correlated with a spectrum of patient traits, the member genes were most often significantly associated with metabolic functions [*Figure 6C*]. However, the strongest enrichments fell within a select set of functions related to cell responses to external stimuli [*Figure 6D*]. This supports the hypothesis that patient phenotypic variance might account for heterogeneous clinical presentation, and that this variance might emerge both in metabolic and signaling functions of the cell. However, the greater enrichment of cell-cell signaling pathways suggest that the phenotype might extend beyond simple metabolic dysfunction and include a breakdown of intercellular communication systems tightly linked to growth and metabolism of the cell. Several modules were significantly associated with one patient flagged with abnormal Behavior scores. However, we do not consider those modules here because Behavior-specific transcriptional associations are confounded with clone-specific associations.

Of the modules most highly associated with patient traits, the pink4 module of genes appeared to have the highest correlation with DEL patient phenotypes, and was one of the few modules highly associated with poor outcomes on a Language metric [*Figure 6—figure supplement 2*]. The genes most highly associated with this module contain regulators of major developmental cascades in development that could conceivably disrupt gene programs related to both nervous system function and general cellular growth, including Wnt pathway negative regulator WIF1, and Erk1/2 MAPK pathway effector FGF1 [*Figure 6E*]. Interestingly, increased Weight Z-score in this patient cohort was associated with reduced expression of insulin (INS), but increased expression of fetal growth-regulating insulin-like growth factor 2. Although the present study characterizes neural tissue, if generalized insulin-related signaling abnormalities exist in these patients, this disruption would likely be compounded by established alterations in MAPK signaling, an insulin signaling effector mediating cell growth, leading to a high BMI phenotype. Taken together with the observation made about the larger four-module block, the identification of BMI-associated genes within NPCs could be attributable to the fact that some aspects of the gene program modulated by 16p11.2 deletion may represent a generalized phenotype with tissue type-specific effects.

The pink4 module is similarly associated with two functions that could be conceivably impacted by dysregulation of the developing brain. One of these changes, Head Circumference Z-score, suggests a possible link with abnormal activity within NPCs, as head circumference has been associated

with aberrant proliferation in NPCs, and DEL patients have been observed to have decreased head circumference (*Qureshi et al., 2014*; *Vaccarino et al., 2009*). Additionally, the association of this module with language deficits suggest a possible link to more mature circuit assembly and function, coincident with previous reports of language disruption associated within the 16p11.2 deletion phenotype (*Blackmon et al., 2018*; *Kim et al., 2020*). This could arise through a negative correlation with expression of *AMIGO1*, a adhesion molecule expressed throughout the nervous system associated with the development of dendrite outgrowth and the formation of fasciculated fiber tracts (*Zhao et al., 2014*) and dendritic outgrowth (*Chen et al., 2012*). These features are also negatively associated with calcium-activated potassium channel member KCNMB4, which has been previously associated with high-order disruption of neural function through its association with ASD in SNP association studies (*Skafidas et al., 2014*). Together, the correlation between these sets of genes and phenotypes associated with abnormal neurodevelopment invites speculation into the heterogeneous nature of language deficits in patients with 16p11.2 CNVs.

It is surprising that the pink4 module is associated not only with neural function and BMI, but also Age. While it is not possible to pinpoint what aspects of the genetic signature might be attributable to this association, the observed relationship between trait and expression of specific genes (e.g. Weight Z-score and *INS* expression) indicate that the combined correlations are not entirely attributable to normal changes in weight with age. The results of the WGCNA analysis suggest, as a whole, that several gene modules may be associated with a spectrum of clinical outcomes with 16p11.2 microdeletion, that some BMI-associated changes are detected even in a cell type presumably unrelated to BMI (neural progenitors), and that some of these correlations with clinical features involve genes outside of the 16p11.2 microdeletion. While further investigation is needed to untangle the complexity of patient-specific gene programs and generic 16p11.2 deletion effects, this work provides a springboard for further interrogation of hiPSC-derived cell types and features common to 16p11.2 microdeletion relative to features that may arise in individual donors due to other genetic background effects.

Finally, we emphasize the importance of screening clones for cryptically integrated reprogramming plasmids, especially as published methods highlight the lack of integration when using episomal reprogramming vectors. It is likely that these effects are most relevant for reprogramming methods that utilize DNA expression vectors and less likely for RNA-based delivery methods, such as modified-RNA or Sendai virus-based reprogramming. Many of the clones used in this study and available in the repository contain random integration of reprogramming plasmids that introduce strong transcriptional artifacts. The impact of cryptic reprogramming vector integration is well illustrated by the segregation of Int+ and Int- clones into identifiable clusters within the first two PCs in *Figure 4A*. These transcriptional perturbations could emerge from two possible sources: (1) an increase in reprogramming factor protein product brought about by the insertion of transcriptionally active transcripts or (2) a disruptive insertion of integrant into a transcriptionally active locus. We note that one Int+ clone (DEL_4_1) has a silenced insertion that contains integrant DNA [*Figure 3— figure supplement 1*] but did not produce detectable transcript that aligns to the integrated plasmid. Moreover, this Int+ clone has an expression profile that closely resembles those of the Int- clones [*Figure 4D*]. This is consistent with the observed transcriptional effects being primarily related to expression of the reprogramming factors rather than random insertional mutations. The strongest correlation appears to be due to plasmid-based expression of *POU5F1*. This is also suggested in our qRT-PCR data, where *POU5F1* transcripts are not detected in clone DE_4_2, which is most similar in DE gene transcriptional patterns observed in the Int- clones [*Figure 3—figure supplement 1*]. The integration and expression of reprogramming vectors is accompanied by the differential expression of more than 3000 genes. Not surprisingly, the integration-associated DE gene list contains 136 genes previously identified as *POU5F1* transcriptional targets, representing approximately 16% of genes cataloged in three putative *POU5F1* target gene lists (*Sharov et al., 2008*). Given that such cryptic integrations may also exist in other hiPSC lines that have been reprogrammed using non-integrating episomal systems, we provide a customizable RNA-seq analysis pipeline that can detect expression from cryptic plasmids. It is computationally lightweight and can feasibly be performed on a laptop computer. This method provides users with the means to quickly interrogate their data and detect whether similar integration effects should be considered and addressed.

In summary, the collection of 65 human hiPSC clones from 13 donors with either a microdeletion or microduplication at the 16p11.2 chromosomal locus provides an important resource for studying

the early neurodevelopmental alterations that may underlie physiological and behavioral features seen in individuals carrying 16p11.2 CNVs. The data presented herein identify several early transcriptomic network perturbations which may act as priming events for later functional alterations in more mature cell types (*Schafer et al., 2019*). A majority of the available clones have been thoroughly evaluated for reprogramming success, ploidy, SNPs, and the capacity to differentiate down the cortical neural lineage. The clones have also been screened and sorted for cryptic integration of the 'non-integrating' reprogramming vectors, and data from the subset of footprint-free lines provide several novel candidates for future studies of 16p11.2 deletion. The complete demographic and diagnostic information for each fibroblast donor, and instructions for obtaining the cells themselves, are available through SFARI at https://sfari.org/resources/autism-models/ips-cells. The comprehensive information collected from donors makes this hiPSC resource particularly valuable for investigating associations between in vitro phenotypes and clinical diagnoses that may be unique to a given individual or shared among 16p11.2 CNV carriers.

# Materials and methods

## Key resources table

| Reagent type (species) or resource | Designation | Source or reference | Identifiers | Additional information |
|---|---|---|---|---|
| Antibody | Anti-Oct4 (Goat polyclonal) | Abcam | Cat. No. ab27985 RRID:AB_776898 | IF (1:200) |
| Antibody | Anti-Nanog (Goat polyclonal) | R and D Systems | Cat. No. AF1997 RRID:AB_355097 | IF (1:200) |
| Antibody | Anti-TRA-1–60 (Mouse monoclonal) | Abcam | Cat. No. ab16288 RRID:AB_778563 | IF (1:266) |
| Antibody | Anti-TRA-2–49 (Mouse monoclonal) | Developmental Studies Hybridoma Bank | Cat No. TRA-2-49/6E RRID:AB_528073 | IF (1:200) |
| Antibody | Anti-Pax6 (Rabbit polyclonal) | Biolegend | Cat. No. 901301 RRID:AB_2565003 | IF (1:200) |
| Antibody | Mouse anti-NCad (Mouse monoclonal) | BD Biosciences | Cat. No. 610920 RRID:AB_2077527 | IF (1:173) |
| Antibody | Anti-ZO-1 (Mouse monoclonal) | Invitrogen | Cat. No. 33–9100 RRID:AB_2533147 | IF (1:100) |
| Antibody | Goat anti-aPKCζ (Goat polyclonal) | Santa Cruz | Cat. No. sc-216 RRID:AB_2300359 | IF (1:200) |
| Antibody | Anti-Pericentrin (Mouse monoclonal) | Abcam | Cat. No. ab28144 RRID:AB_2160664 | IF (1:200) |
| Antibody | Anti-pHH3 (Goat polyclonal) | Santa Cruz | Cat. No. sc-12927 RRID:AB_2233069 | IF (1:200) |
| Antibody | Anti-Emx1 (Rabbit polyclonal) | Thermo Scientific | Cat. No. PA5-35373 RRID:AB_2552683 | IF (1:100) |
| Antibody | Anti-Dlx1 (Mouse monoclonal) | Abcam | Cat. No. ab54668 RRID:AB_941307 | IF (1:250) |
| Antibody | Anti-Tuj1 (Mouse monoclonal) | Abcam | Cat. No. ab78078 RRID:AB_2256751 | IF (1:200) |

*Continued on next page*

*Continued*

| Reagent type (species) or resource | Designation | Source or reference | Identifiers | Additional information |
|---|---|---|---|---|
| Antibody | Anti-NeuN (Rabbit monoclonal) | Abcam | Cat. No. ab177487 RRID:AB_2532109 | IF (1:300) |
| Antibody | Anti-rabbit conjugated with Alexa-488 (Goat polyclonal) | Invitrogen | Cat. No. A-11034 RRID:AB_2576217 | IF (1:500) |
| Antibody | Anti-mouse conjugated with Alexa-488 (Goat polyclonal) | Invitrogen | Cat. No. A-11001 RRID:AB_2534069 | IF (1:500) |
| Antibody | Anti-rabbit conjugated with Alexa-488 (Donkey polyclonal) | Jackson | Code No. 711-545-152 RRID:AB_2313584 | IF (1:500) |
| Antibody | Anti-goat conjugated with Alexa-488 (Donkey polyclonal) | Jackson | Code No. 705-545-003 RRID:AB_2340428 | IF (1:500) |
| Antibody | Anti-mouse conjugated with Alexa-488 (Donkey polyclonal) | Jackson | Code No. 715-545-150 RRID:AB_2340820 | IF (1:500) |
| Antibody | Anti-rabbit conjugated with Cy3 (Donkey polyclonal) | Jackson | Code No. 711-165-152 RRID:AB_2307443 | IF (1:500) |
| Antibody | Anti-goat conjugated with Cy3 (Donkey polyclonal) | Jackson | Code No. 705-165-147 RRID:AB_2307351 | IF (1:500) |
| Antibody | Anti-mouse conjugated with Cy3 (Donkey polyclonal) | Jackson | Code No. 715-165-150 RRID:AB_2340813 | IF (1:500) |
| Antibody | Anti-rabbit conjugated with Cy5 (Donkey polyclonal) | Jackson | Code No. 711-175-152 RRID:AB_2340607 | IF (1:500) |
| Antibody | Anti-goat conjugated with Cy5 (Donkey polyclonal) | Jackson | Code No. 705-175-147 RRID:AB_2340415 | IF (1:500) |
| Antibody | Anti-mouse conjugated with Cy5 (Donkey polyclonal) | Jackson | Code No. 715-175-151 RRID:AB_2340820 | IF (1:500) |
| Chemicals Compound, Drug | RHO/ROCK Pathway Inhibitor Y27632 | Stem Cell Technologies | Cat. No. 72302 | |
| Chemicals Compound, Drug | SB-431542 | Tocris | Cat. No. 1614 | |
| Chemicals Compound, Drug | LDN-193189 | Stemgent | Cat. No. 04007402 | |

*Continued*

| Reagent type (species) or resource | Designation | Source or reference | Identifiers | Additional information |
|---|---|---|---|---|
| Chemicals Compound, Drug | ProLong Gold Antifade Mountant with DAPI | Thermo Fisher Scientific | Cat. No. P36931 | |
| Chemicals Compound, Drug | TRIzol | Thermo Fisher Scientific | Cat. No. 15596026 | |
| Peptides, Recombinant Proteins | Human Recombinant FGF2 | R and D Systems | Cat. No. 233-FB | |
| Peptides, Recombinant Proteins | hESC-qualified Matrigel Matrix | Corning | Product No. 354277 | |
| Peptides, Recombinant Proteins | Human Recombinant Insulin | Thermo Fisher Scientific | Cat. No. 12585014 | |
| Peptides, Recombinant Proteins | Poly-D-Lysine | Sigma Aldrich | Cat. No. P7280 | |
| Peptides, Recombinant Proteins | Laminin | Roche | Cat. No. 11243217001 | |
| Commercial Assay, Kit | Amaxa Human Dermal Fibroblast Nucleofector Kit | Lonza | Cat. No. VDP - 1001 | |
| Commercial Assay, Kit | Epi5 Reprogramming Kit | Thermo Fisher Scientific | Cat. No. A15960 | |
| Commercial Assay, Kit | GeneJET Genomic DNA purification KIT | Thermo Fisher Scientific | Cat. No. K0722 | |
| Commercial Assay, Kit | Platinum Taq DNA Polymerase High Fidelity Kit | Thermo Fisher Scientific | Cat. No. 11304011 | |
| Commercial Assay, Kit | GeneJET RNA purification kit | Thermo Fisher Scientific | Cat. No. K0731 | |
| Commercial Assay, Kit | High Capacity cDNA Reverse Transcription Kit | Applied Biosystems | Cat. No. 4368814 | |
| Cell Line (*Homo-sapiens*) | 16p11.2 Deletion and Duplication iPSC lines | SFARI | See *Figure 2— figure supplement 1* | |
| Cell Line (*M. musculus*) | DR4 MEF Feeder Cells | Transgenic Mouse Facility at Stanford University | | |
| Recombinant DNA Reagent | pCXLE-hOCT3/4-shp53-F | Addgene | Cat. No. Plasmid 27077 | |
| Recombinant DNA Reagent | pCXLE-hSK | Addgene | Cat. No. Plasmid 27078 | |
| Recombinant DNA Reagent | pCXLE-hUL | Addgene | Cat. No. Plasmid 27080 | |
| Sequence-based Reagent | pCXLE-hOCT4-shp53-F-fwd | This paper | PCR primers | CAGTGTCCTTT CCTCTGGCCCC |

*Continued on next page*

*Continued*

| Reagent type (species) or resource | Designation | Source or reference | Identifiers | Additional information |
|---|---|---|---|---|
| Sequence-based Reagent | pCXLE-hOCT4-shp53-F-rev | This paper | PCR primers | ATGAAAGCCAT ACGGGAAGC AATAGC |
| Sequence-based Reagent | pCXLE-hSK-fwd | This paper | PCR primers | AATGCGACCGAG CATTTTCCAGG |
| Sequence-based Reagent | pCXLE-hSK-rev | This paper | PCR primers | TGCGTCAGCAAAC ACAGTGCACA |
| Sequence-based Reagent | pCXLE-hUL-fwd | This paper | PCR primers | CAGAGCATCAGCC ATATGGTAGCCT |
| Sequence-based Reagent | pCXLE-hUL-rev | This paper | PCR primers | ACAACGGGCC ACAACTCCTCAT |
| Sequence-based Reagent | Actb-fwd | This paper | PCR primers | AGAGCTACGAG CTGCCTGAC |
| Sequence-based Reagent | Actb-rev | This paper | PCR primers | AGCACTGTGT TGGCGTAGAC |
| Sequence-based Reagent | Hand1-fwd | This paper | PCR primers | GTGCGTCCTTT AATCCTCTTC |
| Sequence-based Reagent | Hand1-rev | This paper | PCR primers | GTGAGAGCA AGCGGAAAAG |
| Sequence-based Reagent | Sox17-fwd | This paper | PCR primers | CGCACGGAAT TTGAACAGTA |
| Sequence-based Reagent | Sox17-rev | This paper | PCR primers | GGATCAGGG ACCTGTCACAC |
| Sequence-based Reagent | Pax6-fwd | This paper | PCR primers | TGGGCAGGTA TTACGAGCTG |
| Sequence-based Reagent | Pax6-rev | This paper | PCR primers | ACTCCCGCTTA TACTGGGCTA |
| Software, Algorithm | SnapGene | SnapGene | RRID:SCR_015052 | |
| Software, Algorithm | ImageJ | ImageJ | RRID:SCR_003070 | |
| Software, Algorithm | Photoshop | Adobe | RRID:SCR_014199 | |
| Software, Algorithm | Prism v7.04 | GraphPad | RRID:SCR_002798 | |
| Software, Algorithm | FastQC v0.11.6 | Babraham Bioinformatics | RRID:SCR_014583 | |
| Software, Algorithm | kallisto v0.43.1 | Pachter Lab | RRID:SCR_016582 | |
| Software, Algorithm | R | R Project for Statistical Computing | RRID:SCR_001905 | |

*Continued*

| Reagent type (species) or resource | Designation | Source or reference | Identifiers | Additional information |
|---|---|---|---|---|
| Software, Algorithm | RStudio | RStudio | RRID:SCR_000432 | |
| Software, Algorithm | Tximport v1.8.0 | tximport | None yet available | |
| Software, Algorithm | DESeq2 v1.20.0 | DESeq2 | RRID:SCR_015687 | |
| Software, Algorithm | ggplot2 v3.0.0 | ggplot2 | RRID:SCR_014601 | |
| Software, Algorithm | pheatmap v1.0.10 | pheatmap | RRID:SCR_016418 | |
| Software, Algorithm | GSEA | Broad Institute | RRID:SCR_003199 | |
| Software, Algorithm | EnrichmentMap | Bader Lab | RRID:SCR_016052 | |
| Software, Algorithm | Cytoscape | Institute for Systems Biology; Washington; USA; University of California at San Diego; California; USA | RRID:SCR_003032 | |
| Software, Algorithm | UCSC Genome Browser | University of California at Santa Cruz; California; USA | RRID:SCR_005780 | |
| Software, Algorithm | STAR v2.5.3a | STAR | RRID:SCR_015899 | |
| Software, Algorithm | LIMMA v3.36.5 | LIMMA | RRID:SCR_010943 | |
| Software, Algorithm | WGCNA | University of California at Los Angeles; California; USA | RRID:SCR_003302 | |
| Software, Algorithm | 16 p resource Code | Kristin L. Muench | RRID:SCR_016845 | |
| Other | Genome-Wide Human SNP Array, 6.0 platform | Affymetrix | Performed by CapitalBio Corp., Beijing, China | |
| Other | Ultra-low attachment and ultra-low cluster 96-well plates | Corning | Cat. No. CLS3474 | |
| Other | Lumox 50 mm plates | Sarstedt | Cat. No. 833925 | |
| Other | NextSeq 500 | Illumina | Performed by Stanford Functional Genomics Facility, Stanford, California, U.S.A. | |

## Human hiPSC generation with episomal vectors

The majority of hiPSC available through SFARI were reprogrammed from skin fibroblasts using episomal vectors encoding *SOX2, OCT3/4, KLF4, LIN28, L-MYC,* and *P53*-shRNA (*Okita et al., 2011*). These vectors are pCXLE-hOCT3/4-shp53-F (Addgene, Watertwon, Massachusetts; Plasmid

27077), pCXLE-hSK (Addgene Plasmid 27078), and pCXLE-hUL (Addgene Plasmid 27080). 6 µg of total plasmid DNA (2 µg for each of the three episomal vectors) were electroporated into $5 \times 10^5$ fibroblasts with the Amaxa Human Dermal Fibroblast Nucleofector Kit (Lonza, Basel, Switzerland; VDP-1001). The fibroblasts were further cultured for 6 days to allow them to recover, and then plated onto mouse DR4 MEF (provided by The Transgenic Mouse Facility at Stanford University). The cells were maintained for 24 hr in growth medium consisting of DMEM/F12 (Thermo Fisher Scientific, Waltham, Massachusetts; 11320033), Fetal Bovine Serum (10%, Thermo Fisher Scientific, 16000044), MEM Non-Essential Amino Acids (NEAA) (1%, Thermo Fisher Scientific, 11140050), Sodium Pyruvate (1%, Thermo Fisher Scientific, 11360070), GlutaMax (0.5%, Thermo Fisher Scientific, 35050061), PenStrep (100 units/mL, Thermo Fisher Scientific, 15140122), and β-mercaptoethanol (55 µM, Sigma Aldrich, M6250). After 24 hr, the culture medium was switched to standard human embryonic stem cell (hESC) medium containing DMEM/F12 (Thermo Fisher Scientific, 11320033), KnockOut Serum Replacement (20%, Thermo Fisher Scientific, 10828028), GlutaMax (0.5%, Thermo Fisher Scientific, 35050061), MEM NEAA (1%, Thermo Fisher Scientific, 11140050), Penstrep (100 units/mL, Thermo Fisher Scientific, 15140122), β-mercaptoethanol (55 µM, Sigma Aldrich, St. Louis, Missouri; M6250), and human recombinant FGF2 (10 ng/ml, R and D Systems, Minneapolis, Minnesota; 233-FB). After 2–3 weeks, emerging hiPSC colonies were manually selected based on morphological standards and expanded clonally into hESC-qualified Matrigel Matrix (0.10 mg/mL, Corning, Corning, New York; 354277) coated plates in mTeSR one medium (Stem Cell Technologies, Vancouver, Canada; 5851). A subset of the available clones was reprogrammed using the Epi5 Reprogramming Kit (Thermo Fisher Scientific, A15960) according to the kit's instructions. The hiPSCs were routinely passaged with Accutase (Stem Cell Technologies, 07920) when they reached high confluency and re-plated in mTeSR one with RHO/ROCK Pathway Inhibitor Y27632 (Stem Cell Technologies, 72302). Patient donors are identified with a code with one numeral (DEL_1) whereas different lines from the same patient are identified with an extended code (e.g. DEL_1_2, DEL_1_3).

Wild-type control lines were all episomally reprogrammed in parallel with microdeletion and microduplication lines. Donor age and sex are as follows: WT_8343 (Female, 5 years), WT_2242 (Male, 4 years), WT_NIH511 (Male, 27 years), and WT_NIH2788 (Female, 30 years). All wild-type lines are available by request from the Stanford Neuroscience iPSC Core.

16p11.2 DEL and DUP iPSC lines were acquired from the Simons Foundation SFARI VIP Collection. All lines tested as mycoplasma-free and donor origin for each iPSC clone was confirmed by SNP analysis, PCR, and RNAseq.

## Expression of pluripotency markers in hiPSCs

All cells maintained in monolayer cultures were fixed with 4% paraformaldehyde for 10 min at room temperature. The pluripotency of the derived hiPSCs was tested by immunocytochemical staining for OCT4 (Abcam, Cambridge, United Kingdom; ab27985, RRID:AB_776898), NANOG (R and D Systems, AF1997, RRID:AB_355097), TRA-1–60 (Abcam, ab16288, RRID:AB_778563) and TRA-2–49 (Developmental Studies Hybridoma Bank, TRA-2-49/6E, RRID:AB_528071). The secondary antibodies were goat anti-rabbit IgG conjugated with Alexa-488 (Invitrogen, Carlsbad, California; A-11034, RRID:AB_2576217) and goat anti-mouse IgG conjugated with Alexa-488 (Invitrogen, A-11011, RRID: AB_2534069).

## Episomal vector integration and transcripts detection

Genomic DNA was extracted from cell pellets using the GeneJET Genomic DNA purification KIT (Thermo Fisher Scientific, K0722) according to supplier protocols. We designed primers using SnapGene software (RRID:SCR_015052) and verified them for the absence of non-specific binding using NCBI Primer-BLAST. Primers were designed to amplify products that spanned part of either one of the reprogramming genes *OCT4, KLF4, LIN28,* and the flanking WPRE region to prevent primer hybridization and amplification of these endogenous genes in the genomic DNA. To assess the sensitivity of the PCR reaction, plasmid DNA (each of the three plasmids separately) was mixed in with H9 genomic DNA to create a dilution curve at concentrations starting at 2000:1 and reducing by factor of ten to a 2:1 copy number of plasmid to genomic DNA. All PCR reactions were carried out using the Platinum Taq DNA Polymerase High Fidelity kit (Thermo Fisher Scientific, 11304011) according to supplier protocols. 1 µl of genomic DNA (approx. 15 ng) was added to the

amplification master mix containing 60 mM Tris-(SO4), 18 mM (NH4)$_2$SO$_4$, 50 mM MgSO$_4$, 10 mM dNTPs, 5 U/µl of Platinum Taq DNA Polymerase, and the forward and reverse primers. The amplification reaction was carried out using a cycling program of 1 min at 95℃, followed by 35 cycles of 15 s at 95℃, 30 s at the annealing temperature specific for each primer set, and 1 min at 68℃ in a Bio-Rad C1000 thermal cycler. Total RNA was extracted from cell pellets using the GeneJET RNA purification kit (Thermo Fisher Scientific, K0731) and cDNA synthesis performed was performed using the High-Capacity cDNA Reverse Transcription kit (Applied Biosystems, Foster City, California; 4368814) according to supplier protocols. For the cDNA synthesis, 3 µl of RNA was added to the master mix (10x RT buffer, 100 mM dNTPs, Multiscribe Reverse Transcriptase, 10X primers), and the reverse transcription reaction was performed using a cycling program of 10 min at 25℃, 120 min at 37℃, and 5 min at 85℃. To detect the presence of vector transcripts, 1 µl of cDNA from each clone was used in an amplification reaction with the same set of primers that were used for the detection of vector integration in gDNA. Primer sequences for integrated plasmids and transcripts were as follows: pCXLE-hOCT4-shp53-F (fwd, CAGTGTCCTTTCCTCTGGCCCC; rev, ATGAAAGCCATACGG-GAAGCAATAGC; 329 bp product), pCXLE-hSK (fwd, AATGCGACCGAGCATTTTCCAGG; rev, TGCGTCAGCAAACACAGTGCACA; 342 bp), pCXLE-hUL (fwd, CAGAGCATCAGCCATATGG TAGCCT; rev, ACAACGGGCCACAACTCCTCAT; 380 bp). H9 hESC DNA was used as negative controls in all amplification reactions.

## Single-nucleotide polymorphisms and copy number variants analyses

The Affymetrix Genome-Wide Human SNP Array 6.0 platform was chosen for SNP and CNV analysis. The SNP array assay was performed and analyzed by CapitalBio Corp., Beijing, China. Genetic markers, including more than 906,600 SNPs and more than 946,000 CNV probes, were included for the detection of both known and novel CNVs. Data were analyzed by a copy number polymorphism (CNP) calling algorithm developed by the Broad Institute (Cambridge, Massachusetts, USA). In addition, sample mismatch analysis was performed with genotyping data of the SNP array to confirm that each clone from a given donor was identical and carried the donor's original genotype (*Simmons et al., 2011*; *Zhu et al., 2011*).

## Differentiation of human hiPSCs to the neural lineage

As previously reported (*Shi et al., 2012*), hiPSCs were differentiated in N3 culture medium which consists of DMEM/F12 (1x, Thermo Fisher Scientific, 11320033), Neurobasal (1x, Thermo Fisher Scientific, 21103049), N-2 Supplement (1%, Thermo Fisher Scientific, 17502048), B-27 Supplement (2%, Thermo Fisher Scientific, 17504044), GlutaMax (1%, Thermo Fisher Scientific, 35050061), MEM NEAA (1%, Thermo Fisher Scientific, 11140050), and human recombinant insulin (2.5 µg/mL, Thermo Fisher Scientific, 12585014). From Day 1 to 11, the N3 media was further supplemented with two factors: SB-431542 (5 µM, Tocris, 1614) and LDN-193189 (100 nM, Stemgent, 04007402). At Day 12, the differentiating cells were dissociated with Cell Dissociation Solution (1x, Sigma-Aldrich, C5914), passaged onto Poly-D-Lysine (50 µg/mL, Sigma-Aldrich, P1024) and Laminin (5 µg/mL, Roche, Basel, Switzerland; 11243217001) coated plates, and cultured in N3 media without factors until Day 22 when they were passaged again. Between Day one and Day 22, media changes were performed daily. However, after Day 22, cells were exposed to media changes every other day with N3 media without factors.

## Differentiation of human hiPSCs to endoderm and mesoderm lineages

hiPSC cells grow in mTeSR were dissociated into single cells using accutase and plated at density of 25–50 k cells/cm$^2$ on matrigel coated cell culture plates and subsequently differentiated into endoderm and mesoderm lineages (*Loh et al., 2014*; *Ang et al., 2018*). For definitive endoderm induction, anterior primitive streak was first specified using 100 ng/ml Activin A (R and D systems, 338-AC-050), 3 µm CHIR (Tocris, 4423) and 20 ng/ml FGF2 (R and D Systems, 233-FB-01M) in CDM2 basal media. After 24 hr, the cells were washed with DMEM/F12 (1x, Thermo Fisher Scientific, 11320033), and definitive endoderm was induced using 100 ng/ml Activin A and 250 nM LDN (Reprocell, Yokohama, Japan; 04–0074) in CDM2 basal media for 24 hr. For lateral mesoderm induction, midprimitive streak was specified using 30 ng/ml Activin, 16 µM CHIR, 20 ng/ml FGF2 and 40 ng/ml BMP (R and D Systems, 314 BP-050) in CDM2 basal media for 24 hr. After 24 hr, cells were

washed with DMEM/F12 (1x, Thermo Fisher Scientific, 11320033), and lateral mesoderm was induced using 1 µM A8301 (R and D Systems, 2939) 30 ng/ml BMP and 1 µM C59 (Tocris, Bristol, United Kingdom; 5148) for 24 hr in CDM2 basal media. On the third day, cells were lysed for RNA collection and purification.

## RNA extraction, reverse transcription, and qPCR

RNA was collected from adherent wells grown in individual wells of a 24-well cell culture plate using the Qiagen RNeasy kit as per the manufacturer's instructions with an added intermediate step of on-column DNA digestion to remove genomic DNA. About 10–100 ng of RNA was used for reverse transcription (High capacity cDNA reverse transcription kit, Applied Biosystems, 4368814) as per the kit manufacturer's instructions. cDNA was then diluted 1:10 and was used for each qPCR reaction in a 384 well format. To assess gene expression of endoderm, mesoderm and ectoderm lineages, in each individual qPCR reaction, 5 µl of (2x) SYBR green master mix (SensiFAST SYBR kit, Bioline, London, United Kingdom; BIO- 94005) was used and combined with 0.4 µl of a combined forward and reverse primer mix (10 µM of forward and reverse primers in the combined master mix) and the reaction was run at Tm of 60℃ for 40 cycles. qPCR analysis was conducted by the ΔΔCt method and the expression of each gene was internally normalized to the expression of a house keeping gene (ACTB) for the same cDNA sample. For all differentiated cells, expression of the lateral mesoderm marker (HAND1), definitive endoderm marker (SOX17), neuroectoderm marker (PAX6) and pluripotency marker (Nanog) was compared to the expression of these markers from the same undifferentiated cell line. The primer sequences for the lineage markers are as follows, Actb (fwd: AGAGCTACGAGCTGCCTGAC, rev: AGCACTGTGTTGGCGTAGAC), Hand1 (fwd: GTGCGTCCTTTAATCCTCTTC, rev: GTGAGAGCAAGCGGAAAAG), Sox17 (fwd: CGCACGGAATTTGAACAGTA, rev: GGATCAGGGACCTGTCACAC), Pax6 (fwd: TGGGCAGGTATTACGAGCTG, rev: ACTCCCGCTTATACTGGGCTA). For the validation of the DESeq interval genes, RNA from cells differentiated for 22 days in vitro into neuroectoderm and subsequently patterned to a dorsal telencephalic identity was used for reverse transcription and qPCR. Genes with largest log2 fold change and most significant p values from the DESeq analysis (TAOK2, Taqman assay Hs00191170_m1; SEZ6L2, Taqman assay Hs03405581_m1; MAPK3, Taqman assay Hs00385075_m1; KCTD13, Taqman assay Hs00923251_m1) compared to wild-type lines were selected. To assess the expression of the 16p11.2 interval genes, in each individual reaction, 5 µl of (2x) Taqman fast advanced master mix was combined with 0.5 µl of the appropriate Taqman gene expression assay probe. qPCR analysis was conducted by the ΔΔCt method and the expression of each gene was internally normalized to the expression of a house keeping gene (Actb) for the same cDNA sample. The expression of the assayed genes was compared to the average expression of 6 wild type hiPSC clones.

## Expression of neural rosette and immature neuronal markers in monolayer cultures

Day 26 differentiated neuroepithelial cells were evaluated via immunocytochemical staining of Pax6 (Biolegend, San Diego, CA; 901301, RRID:AB_2565003), NCad (BD Biosciences, San Jose, California; 610920, RRID:AB_2077527), ZO-1 (InVitrogen, 33–9100, RRID:AB_2533147), aPKCζ (Santa Cruz, Dallas, Texas; sc-216, RRID:AB_2300359), Pericentrin (Abcam, ab28144, RRID:AB_2160664), and pHH3 (Santa Cruz, sc-12927, RRID:AB_2233069). Day 45 immature neurons were stained for Tuj1 (Abcam, ab78078, RRID:AB_2256751) and NeuN (Abcam, ab177487, RRID:AB_2532109). The secondary antibodies were donkey anti-rabbit IgG conjugated with Alexa-488 (Jackson, Bar Harbor, Maine; 711545152, RRID:AB_2313584), donkey anti-mouse IgG conjugated with Cy3 (Jackson, 715165150, RRID:AB_2340813), and donkey anti-goat IgG conjugated with Cy5 (Jackson, 705175147, RRID:AB_2340415). Quantification of pHH3 was performed using ImageJ (RRID:SCR_003070) and Adobe Photoshop (RRID:SCR_014199), while all statistical evaluations were performed with GraphPad Prism v7.04 (RRID:SCR_002798).

## Flow cytometry

Cells were dissociated by treatment with 0.5 mM EDTA (Thermo Fisher Scientific, 15575020) in phosphate-buffered saline (PBS) (Thermo Fisher Scientific, 10010023) for 5 min, or Cell Dissociation Solution (1x, Sigma-Aldrich, C5914) for 20 min, washed with PBS, fixed in 4% paraformaldehyde for 10

min, and washed again with PBS. Cells were then permeabilized with 0.1% Triton X-100 (Thermo Fisher Scientific, 85111) and stained with conjugated primary antibodies for 30 min at manufacturer-recommended concentrations at 4°C in the dark. Antibodies used were anti-Pax6 Alexa 647 (BD Biosciences 562249), and anti-Oct4 Phycoerythrin (PE) (BD Biosciences 560186). Isotype matched controls with corresponding fluorchrome conjugates were acquired from BD Biosciences. Cells were analyzed using a BD Aria II.

## RNA sequencing

Day 22 cortical NPCs were lysed using TRIzol (Thermo Fisher Scientific, 15596026) and stored in −80°C. The samples were then delivered to the Stanford Functional Genomics Facility (SFGF) where they were multiplexed and sequenced using paired end mRNA sequencing on an Illumina NextSeq 500. Raw data were demultiplexed at the sequencing facility. RIN scores for generated libraries were nearly all above 9.0 (range: 7.8–10.0). Reads were 75 basepairs in length, and samples were sequenced with a targeted effective sequencing depth of 40,000,000 reads per sample. Given the high quality of each read, as established by a per-base sequence quality greater than 28 in FastQC (v0.11.6, RRID:SCR_014583), no further read trimming was performed.

## Assessment of the impact of reprogramming factor integration

The. fastq files produced by RNA-seq were aligned using kallisto (v0.43.1) (*Bray et al., 2016*) to a unique index composed of a human reference transcriptome (Ensembl GRCh38.93) and plasmid sequences. The mean number of reads processed per sample was 102,963,875 (range: 65,427,345–193,569,749), and these pseudoaligned to a reference mRNA at a mean rate of 76.34%. By adding plasmid sequences as additional 'chromosomes', we were able to investigate whether sequencing reads aligned to non-genomic as well as genomic DNA. Estimated counts produced by kallisto for each were imported into RStudio for analysis with R using the tximport function (v.1.8.0) (*Soneson et al., 2015*) and imported into a DESeq2 object for further analysis. Genes with one or fewer detected reads across samples were filtered out to expedite computation. Due to the polycistronic nature of the transcripts produced by plasmids, the same expression value for a given plasmid was considered associated with each of its component reprogramming factor genes. PCA plots were produced using the plotPCA function of the DESeq2 (v1.20.0, RRID:SCR_015687) package (*Love et al., 2014*), barplots produced with ggplot2 (v3.0.0, RRID:SCR_014601) (*Kolde and Kolde, 2018*), and heatmaps produced with pheatmap (v1.0.10, RRID:SCR_016418) (*Kolde and Kolde, 2018*). Differential gene expression was calculated using DESeq2 and reported p-values represent FDRs calculated using Benjamini-Hochberg multiple hypothesis testing correction provided by that software pipeline. The threshold for gene significance was set at 0.05. Several batch effect variables were included in the design matrix for the analysis (i.e. patient Sex, patient CNV status, and sequencing day) to ensure that this analysis focused on the transcriptional effects of integration and not a potential batch effect. Pheatmap Z-scores are calculated by subtracting the mean from each input expression value (centering) and dividing by the standard deviation (scaling). To investigate potential biological mechanisms impacted by reprogramming factor integration, the genes were ranked according to a metric representing statistical significance combined with the direction of fold change. Using this scheme, the genes were effectively ranked such that the first and the last entries in the list represented the smallest p-value upregulated and smallest p-value downregulated genes, respectively. The resulting list was submitted to GSEA (RRID:SCR_003199) to identify functional gene modules enriched at the top or the bottom of the list with FDR < 0.25. A subset of the resulting enriched modules was visualized using the EnrichmentMap (RRID:SCR_016052) plugin for Cytoscape (RRID:SCR_003032; *Shannon et al., 2003*). A list of 16p11.2 region genes was acquired by exporting all annotated transcripts in the region from the USCS Genome Browser (RRID:SCR_005780).

## Differential expression analysis

A subset of the. fastq files from the previous analysis corresponding to Int- clones were re-aligned using STAR (v2.5.3a, RRID:SCR_015899). The aligned reads were counted using htseq-count. The mean number of uniquely mapped reads for each sample when imported into R was 83,387,539 reads per sample (range: 54,003,338–164,989,807). Given that this was a paired-end sequencing

experiment, the effective read depth was a mean of 41,693,770 reads per sample. These raw data were batch corrected using SVA (v3.28.0) (*Leek et al., 2012*), and analyzed using DESeq2 for differential expression analysis.DESeq2 uses the given data to model each gene as a negative binomial generalized linear model, and tests for differential expression using the Wald Test and Benjamini-Hochberg adjusted p-values. For visualization of batch effect corrected data, data were transformed using limma (v3.36.5, RRID:SCR_010943) (*Ritchie et al., 2015*). Shrunken log2-fold changes are used according to the recommendation in DESeq2 to account for fold change inflation in low expression genes. Heatmaps are generated using the R package pheatmap, which created a dendrogram of gene similarity using kmeans clustering. Expression values from RNA-Seq data aligned using STAR were also used to visualize fate marker expression in vitro. Gene ontology enrichment analysis was performed using DAVID (RRID:SCR_001881) (*Huang et al., 2009a*; *Huang et al., 2009b*). Gene ontology enrichments were ordered by unadjusted p-values, and the top entries plotted. An indication was added where an enrichment reached statistically significant enrichment following Bonferroni-adjusted p-values.

The validation differential expression method was performed on integration-negative clones using the limma/voom pipeline. In brief, raw count data was normalized with the cpm method, and genes with no detectable expression removed. A model matrix including Genotype, Sex, and GrowBatch as cofactors was fit, and voom() applied to prepare the data for linear modeling by log2-scaling data and estimating the mean-variance relationship. CPM-normalized and batch-effect corrected data indicated that line 8343.5 might be an outlier, but analysis of the weights of the first two principal components revealed that this was due entirely to an abundance of detected X-linked antisense RNA TSIX, and therefore the sample was included in downstream analysis. Next, duplicateCorrelation() applied to account for clone donor through a linear mixed model. A linear model was fit to the data using lmFit(), and DEL vs. WT contrasts calculated using makeContrasts() and contrasts.fit(). Standard error smoothing using eBayes() was performed. Genes were identified as differentially expressed if the Benjamini-Hochberg adjusted p-value fell below 0.05.

## Linking modules of coexpressed genes to DEL patient phenotypes

Modules of gene expression were identified using the WGCNA package (RRID:SCR_003302). The input data to WGCNA were VST-normalized, batch corrected counts from integration-free DEL clones. The default recommendations from WGCNA were used to achieve a relatively scale-free network and unsigned modules correlated of gene modules identified through the blockwiseConsensusModules() function. In brief, the input expression matrix is divided into blocks, and a topological overlap matrix (TOM) created for each block. Clusters are identified from each TOM using average linkage hierarchical clustering, and Dynamic Hybrid tree cut used in to identify preliminary modules. Final modules are identified by merging modules with highly correlated TOMs. Correlations between modules and traits represent Pearson Correlations. Statistically significant associated between modules and traits were modeled using a linear mixed model implemented with the lmer() package with patient used as a grouping factor. The p-value of the association was calculated using a likelihood ratio test implemented by the anova() function with type set to 'Chisq'. Gene module memberships were calculated by assigning genes with the highest correlation to module eigengenes (kMR) within the blockwiseConsensusModules().

Genes assigned by WGCNA to the four modules with most significant patient trait relevance were ranked by their Weight_Z score, with most extreme positive values at the top of the list and submitted to GSEA. The resulting gene program enrichments were submitted to Cytoscape as in *Figure 4E*.

## Distribution of materials

The distribution of hiPSCs is by permission from The Simons Foundation Autism Research Initiative. Additional information on the Variation in Phenotype Project and the process for requesting biospecimens can be found at https://sfari.org/resources/autism-models/ips-cells. Code for bioinformatic analyses in this paper are available at https://www.github.com/kmuench/16p_resource (DOI: 10.5281/zenodo.1948176).

## Acknowledgements

We thank the 16p11.2 CNV donors and their families for their essential contributions to this resource, as well as Sara Walton for her efforts in expanding and cryopreserving of many of the clones received from the SFARI collection. We additionally thank the Stanford Functional Genomics Facility for their assistance in generating the RNA-seq data analyzed in this manuscript. JGR would like to acknowledge the National Science Foundation Graduate Research Fellowship Program and the Stanford Graduate Fellowship. This work was supported by the National Institute of Mental Health (NIMH) grant 1R01MH108660 to TDP.

## Additional information

### Funding

| Funder | Grant reference number | Author |
| --- | --- | --- |
| National Institute of Mental Health | 1R01MH108660 | Theo D Palmer |
| Simons Foundation | SFARI Research Contract | Ricardo E Dolmetsch |

The funders had no role in study design, data collection and interpretation, or the decision to submit the work for publication.

### Author contributions

Julien G Roth, Conceptualization, Data curation, Formal analysis, Investigation, Methodology, Writing - original draft, Writing - review and editing; Kristin L Muench, Conceptualization, Formal analysis, Methodology, Writing - original draft, Writing - review and editing; Aditya Asokan, Formal analysis, Investigation, Methodology; Victoria M Mallett, Data curation, Validation, Investigation, Writing - review and editing; Hui Gai, Data curation, Formal analysis, Investigation, Methodology; Yogendra Verma, Stephen Weber, Carol Charlton, Data curation, Formal analysis, Methodology; Jonas L Fowler, Investigation, Methodology; Kyle M Loh, Formal analysis, Investigation, Methodology, Writing - review and editing; Ricardo E Dolmetsch, Conceptualization, Data curation, Formal analysis, Funding acquisition, Investigation, Methodology, Project administration, Writing - review and editing; Theo D Palmer, Conceptualization, Data curation, Formal analysis, Supervision, Funding acquisition, Validation, Investigation, Visualization, Methodology, Writing - original draft, Project administration, Writing - review and editing

### Author ORCIDs

Julien G Roth https://orcid.org/0000-0002-7560-3258
Ricardo E Dolmetsch http://orcid.org/0000-0002-2738-8338
Theo D Palmer https://orcid.org/0000-0002-6266-1862

### Decision letter and Author response

Decision letter https://doi.org/10.7554/eLife.58178.sa1
Author response https://doi.org/10.7554/eLife.58178.sa2

## Additional files

### Supplementary files

• Supplementary file 1. Demographic, diagnostic, and breakpoint information. Abbreviations: NA, not applicable; ND, not determined.

• Supplementary file 2. Pluripotency, vector silencing, and neural competency information. (A) Subset summary of quality control data for hiPSC reprogramming and differentiation. (B) Entire quality control data for hiPSC reprogramming and differentiation. Abbreviations: NA, not applicable; ND, not determined. Plasmids used for the PCR targeted a portion of the plasmid-specific WPRE region.

- Supplementary file 3. CNVs located outside the 16p11.2 chromosomal locus. SNP array analysis revealed hiPSC-clone-specific CNVs throughout the genome. Their copy number, location, and size are listed.

- Supplementary file 4. Gene sets significantly enriched in ranked list of DESeq2 output Comparing Int+ and Int- Clones. All genes submitted to DESeq2 were ranked according to the -log10 of adjusted p-value and the sign of their fold change, such that the top of the list represented upregulated and significant genes, and the bottom of the list downregulated and significant genes, with non-significant and high p-value genes toward the middle. Gene sets were generated by GSEA and additional annotation provided by Enrichment Map.

- Supplementary file 5. Differentially expressed genes in 16p11.2 deletion clones relative to control clones. Annotated differentially expressed gene list generated by DESeq2. Genes are ranked by adjusted p-value.

- Supplementary file 6. DAVID gene ontology enrichment analysis. Gene ontology enrichments were ordered by unadjusted p-values.

- Supplementary file 7. GSEA output following WGCNA gene identification. Pathways as characterized as upregulated ('na_pos') within the set of genes that make up the modules blue, bisque4, salmon, and pink4.

- Transparent reporting form

## Data availability

RNAseq data GEO Submission GSE144736 All additional data is included in the manuscript and supporting files.

The following dataset was generated:

| Author(s) | Year | Dataset title | Dataset URL | Database and Identifier |
|---|---|---|---|---|
| Palmer TD, Muench K | 2020 | Copy Number Variation at 16p11.2 Imparts Transcriptional Alterations in Neural Development in an hiPSC-derived Model of Corticogenesis | https://www.ncbi.nlm.nih.gov/geo/query/acc.cgi?acc=GSE144736 | NCBI Gene Expression Omnibus, GSE144736 |

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
