## [Decision Letter]

**Acceptance summary:**

This important resource paper describes a collection of induced pluripotent stem cells (iPSCs) produced from patients with a microdeletion of the 16p11.2 chromosomal locus, well-known to be associated with significant neurodevelopmental and neuropsychiatric abnormalities. The work provides both a cautionary tale about some unexpected consequences of the reprogramming process, as well as a plethora of information about changes in gene expression observed when neural progenitors are derived from these iPSCs. The results provided here, combined with the availability of these iPSCs from the Simons Foundation Autism Research Initiative, should be useful to developmental neurobiologists, those interested in psychiatric disease, and, frankly, to anyone interested in large-scale iPSC-based disease modeling.

**Decision letter after peer review:**

Thank you for submitting your article "16p11.2 CNV Imparts Transcriptional Alterations in Neural Development in an hiPSC Model of Corticogenesis" for consideration by *eLife*. Your article has been reviewed by two peer reviewers, and the evaluation has been overseen by a Reviewing Editor and Marianne Bronner as the Senior Editor. The reviewers have opted to remain anonymous.

The reviewers have discussed the reviews with one another and the Reviewing Editor has drafted this decision to help you prepare a revised submission.

Summary:

The authors produced 65 iPSC clones from 13 individuals with CNVs at the 16p11.2 locus – 10 with microdeletions and 3 with microduplications. This locus is well-known to be associated with neurodevelopmental and neuropsychiatric abnormalities. The donor cells were provided by the Simons Foundation who also will make the iPSC clones available, making this a valuable cellular resource for those interested in this chromosomal locus and its role in brain development and function. The authors carried out a detailed analysis of the iPSC clones, produced using episomal reprogramming vectors, finding that many of the clones were compromised by unexpected, random integration of the reprogramming vectors with consequent effects on gene transcription in derived cells. The production of this large set of iPSC clones, as well as the cautionary tale concerning vector integration and a detailed RNAseq analysis of the integration-free clones make this paper, as a Tools and Resource publication, a valuable resource for the community.

We are mindful that additional experiments cannot be readily done at the moment. However, we believe that addressing some of these points, together with a reanalysis of the current data, will make this paper more useful to those interested in this area:

1) The data in this paper are restricted to iPSCs generated by episomal reprogramming. Since many, perhaps most, investigators are using Sendai virus, the authors should mention that point. Also do they conclude that episomal reprogramming should be avoided?

2) The analysis uses each clone a separate entity. As best as the reviewers can judge, 6 affected donors, not all of whom were diagnosed with ASD, 3 of which had 3 clones, 1 of which had 2 and the other 2 had 1 each, were used. There were also 3 unaffected donors (with 3, 2, and 2 clones). However, clones should not be weighted the same as individual donors. From Hoffman et al. Nat Comm 2017: "Due to the "repeated measures" study design where individuals are represented by multiple independent hiPSC-NPC and hiPSC-neuron lines, we used a linear mixed model by applying the duplicateCorrelation function in our limma/voom analysis. This approach is widely used to control the false positive rate in studies of repeated measures and its importance in hiPSC data sets was recently emphasized (citing Germain and Testa SCR 2017)". We recommend that the authors reanalyze their data so that each entry is a patient-associated value, rather than a clone-associated one.

3) Based on the results shown in this paper, it is a bit misleading to "strongly suggest" that "16p11.2 deletion may impact many functions important to neurodevelopment and neuropsychiatric disease" – given that the transcriptional changes reported are quite limited and that no data of any observable phenotype was presented. It would be appropriate to tone down that statement and draw more conservative conclusions.

4) Unless the authors wish to include data for the duplication clones, The title "16p11.2 CNV imparts…" should be "16p11.2 microdeletion imparts…" since all duplication hiPSC clones in the analysis were excluded in the analysis steps of Figure 5 and 6.

5) QC questions:

a) Information on the controls is lacking in the main manuscript. Could the authors specifically clarify (in the Results or Materials and methods) the extent to which cases and controls are sex-, ethnicity- and age-matched, as well as whether they were reprogrammed together with the cases and by the same methods?

b) What was the polygenic risk score for the patients/controls? Might this have accounted for variable clinical presentations?

c) Were the hiPSCs karyotyped?

d) What QC was performed on RNAseq to confirm sample identity, cell type composition, etc? It would be helpful if the authors applied variance partition to uncover to what extent variation is explained by reprogramming method, genotype, donor, diagnosis, cell type, etc?

e) What is the Spearman's correlation coefficients for transcriptomic gene expression between donors and clones samples?

6) Questions involving additional useful data analysis

a) Integration with existing datasets and psychiatric genetic studies: Is there an enrichment of 16p11.2 DEGs for the GWAS loci for ASD, SZ or ADHD? Can the authors also test for enrichment of postmortem disease signatures as reported by the CommonMind Consortium, NIH HBCC and/or PsychEncode UCLA (Geschwind) analyses?

b) Are gene expression differences validated in existing 16p11.2 postmortem, hiPSC or mouse datasets (ie. PMID: 26829649)?

c) Given some recent discussions of convergence between CNV effects in psychiatric disease, are major findings from 16p11.2 conserved in hiPSC neuron datasets from other major CNVs (ie. 22q11.2 hiPSC PMID: 27846841 and mouse PMID: 29666363; 15q11.2 BioRxiv https://doi.org/10.1101/772541; NRXN1 PMID: 31784728)

d) Can the authors discuss whether they believe the genome-wide transcriptomic effects of 16p11.2 reflect chromatin loops between the CNV locus and other target genes? miR targets? Or downstream impacts of perturbed neuronal maturation/activity/signaling?

7) Are the authors able to comment on any phenotypic differences that they did observe between microdeletion lines and wt lines?

Revisions expected in follow-up work:

1) Characterization of micro-duplication lines.

2) Elaboration of phenotypic differences at the neural progenitor stage or in differentiated neurons.

---

## [Author Response]

Revisions for this paper:We are mindful that additional experiments cannot be readily done at the moment. However, we believe that addressing some of these points, together with a reanalysis of the current data, will make this paper more useful to those interested in this area:1) The data in this paper are restricted to iPSCs generated by episomal reprogramming. Since many, perhaps most, investigators are using Sendai virus, the authors should mention that point. Also do they conclude that episomal reprogramming should be avoided?

Although we observe widespread integration of the reprogramming vector, we do not conclude that episomal reprogramming should be avoided. In fact, we maintain that episomal reprogramming should be the preferred method of reprogramming over using an integrating vector, such as AAV or Lentiviral vectors. Our observations emphasize that it is important to check for vector integration when the reprogramming factors are introduced as DNA. Although not directly addressed in the current manuscript, we have observed that some disease-relevant mutations appear to greatly reduce the efficiency of reprogramming, for example, LRRK2 and SNCA alterations in Parkinson’s disease, as well as CHD8 and 16p11.2 microdeletion in autism. Our suspicion is that rare random integration provides prolonged expression of the reprogramming factors and confers a selective growth advantage and increases the likelihood that selected clones will carry cryptic integrations.

Manuscript edit:

“Finally, we would like to emphasize the importance of screening clones for cryptically integrated reprogramming plasmids, even though published methods highlight the lack of integration when using episomal reprogramming vectors. It is likely that these effects are most relevant for reprogramming methods that utilize DNA expression vectors and less likely for RNA-based delivery methods, such as modified-RNA or Sendai virus reprogramming.”

2) The analysis uses each clone a separate entity. As best as the reviewers can judge, 6 affected donors, not all of whom were diagnosed with ASD, 3 of which had 3 clones, 1 of which had 2 and the other 2 had 1 each, were used. There were also 3 unaffected donors (with 3, 2, and 2 clones). However, clones should not be weighted the same as individual donors. From Hoffman et al. Nat Comm 2017: "Due to the "repeated measures" study design where individuals are represented by multiple independent hiPSC-NPC and hiPSC-neuron lines, we used a linear mixed model by applying the duplicateCorrelation function in our limma/voom analysis. This approach is widely used to control the false positive rate in studies of repeated measures and its importance in hiPSC data sets was recently emphasized (citing Germain and Testa SCR 2017)". We recommend that the authors reanalyze their data so that each entry is a patient-associated value, rather than a clone-associated one.

We sincerely thank the reviewers for suggesting the use of a linear mixed model to analyze significance in our dataset. Inspired by this point of feedback, we have not only completed a full re-analysis of our differential expression results in our integration-free clones, as described in the cited Hoffman et al., but have also implemented a linear mixed model when testing for the association trait-module association in the WGCNA analysis added in new Figure 6 and Figure 5—figure supplement 4.

In the re-analysis of our integration-free clones, our main findings remain intact. Using the limma/voom DuplicateCorrelation() method, we observe that ten 16p11.2 region genes have a statistically significant reduction in expression. Additionally, we find many of the same genes discussed in the text are differentially expressed, including MIR6723. We have included the results of this new analysis in the supplement as a valuable orthogonal approach from which to interpret this data.

Manuscript edit:

“We further validated these results by identifying DE genes using an orthogonal approach. Using a linear mixed model implemented by the DuplicateCorrelations() function of limma/voom to account for shared patient identities across clones, we identified 40 genes as DE, 17 of which were in common with the genes presented in the first differential expression analysis (Figure 5—figure supplement 4A). […] Taken together, these results strengthen the interpretation of this data as early transcriptional dysregulation with impacts that could affect later gene development.”

Finally, we would like to note that we have included our original analysis in the manuscript (revised Figure 5), which accounted for patient effects through the use of surrogate variables. Surrogate variables can be used to identify patterns of variation not attributable to known experimental design; here, we observed that the magnitude of the loading in our calculated surrogate variables corresponded well to patient. Here, the X axis indicates patient, while the Y-axis indicates magnitude of surrogate variable loading for each sample. The clones from each donor show strong co-variance across the 3 most significant surrogate variables (SV1-3). See Author response image 1.

3) Based on the results shown in this paper, it is a bit misleading to "strongly suggest" that "16p11.2 deletion may impact many functions important to neurodevelopment and neuropsychiatric disease" given that the transcriptional changes reported are quite limited and that no data of any observable phenotype was presented. It would be appropriate to tone down that statement and draw more conservative conclusions.

Numerous wording changes have been integrated throughout the manuscript to avoid overstating the significance of the observations.

4) Unless the authors wish to include data for the duplication clones, The title "16p11.2 CNV imparts…" should be "16p11.2 microdeletion imparts…" since all duplication hiPSC clones in the analysis were excluded in the analysis steps of Figure 5 and 6.

We have edited the title of our manuscript to:

“16p11.2 Microdeletion Imparts Transcriptional Alterations in Human iPSC-derived Models of Early Neural Development”

We have also made text edits throughout the manuscript to reflect our primary focus on microdeletion clones.

5) QC questions:a) Information on the controls is lacking in the main manuscript. Could the authors specifically clarify (in the Results or Materials and methods) the extent to which cases and controls are sex-, ethnicity- and age-matched, as well as whether they were reprogrammed together with the cases and by the same methods?

Demographic data for the control lines is now also included in the Materials and methods section of the manuscript. All cell lines used within this study were reprogrammed in the same lab with the same reprogramming method.

Text edits:

“Wild-type control lines were all episomally reprogrammed and are age and sex matched. Specific demographic information for these lines is listed subsequently: WT_8343 (Female, 5yrs), WT_2242 (Male, 4yrs), WT_NIH511 (Male, 27 yrs), and WT_NIH2788 (Female, 30yrs). All wild-type lines are by request from the Stanford Neuroscience iPSC Core (https://neuroscience.stanford.edu/research/programs/community-labs/neuroscienceipsc-core).”

b) What was the polygenic risk score for the patients/controls? Might this have accounted for variable clinical presentations?

This is an excellent question but we do not have scores for the control lines, and it is not possible to compare patient vs. controls at the present time.

c) Were the hiPSCs karyotyped?

We used SNP arrays to document changes in copy number rather than karyotype analysis. With the exception of the copy number variants noted, all lines are diploid with no missing or extra chromosomes.

d) What QC was performed on RNAseq to confirm sample identity, cell type composition, etc? It would be helpful if the authors applied variance partition to uncover to what extent variation is explained by reprogramming method, genotype, donor, diagnosis, cell type, etc?

The high quality of the initial input.fastqc files was established using MultiQC.

Further QC was performed predominantly through observation of principal components analysis. In VST-normalized data before batch correction, we observe that 50% of the variance is captured within the first two principal components, and 82% of the variance explained by the first ten components. The variance captured by each principal component is depicted as a scree plot (see Author response image 2). The principal components were calculated using the top 500 most highly variable genes as input, as is the method of the DESeq2 package.

**Author response image 2. respfig2:** 

It is clear that much of the variance in the first two components is being driven by the integration phenotype, indicated by the non-overlap of the relative distributions in the density plot depicted on the diagonal of this figure (Author response image 3). Here, the diagonal represents a density plot of subject-wise loading of principal components, the rows and columns represent principal components, and each point represents a clone.

**Author response image 3. respfig3:** 

Despite this, it is clear that there is some separation by Genotype within the first few components, but this is overwhelmed by large variance in the DEL clones, which were highly affected by the integration phenotype (see Author response image 4).

**Author response image 4. respfig4:** 

There is also detectable separation by Sex, GrowBatch, and SubjectID (donor), depicted here (see Author response image 5 and Author response image 6), although they do not appear to be the dominant sources of variation within the first ten principal components.

**Author response image 5. respfig5:** 

**Author response image 6. respfig6:** 

Taken in combination with the correlation between known potential batch effects and surrogate variable analysis, these results help motivate the selection of sex and cell line thaw and grow batch as batch effects in our subsequent analysis. A second confirmatory PCA analysis guided our selection of batch effects in the integration-free clones.As a final confirmation of the of sample identity, sex and patient 16p11.2 status could be identified from the relative expression of sex-linked genes and 16p11.2-region genes, respectively.

e) What is the Spearman's correlation coefficients for transcriptomic gene expression between donors and clones samples?

We have included in Author response image 7 a heatmap of the Spearman’s correlation coefficient for correlation between VST-normalized, batch-corrected expression between samples. The data clusters best by Genotype. Samples are more likely to cluster by Line (representing clones) than by SubjectID (representing clones). In general, this suggests to us that the primary signal in our differential expression analysis will represent differences attributable to Genotype.

**Author response image 7. respfig7:** 

6) Questions involving additional useful data analysisa) Integration with existing datasets and psychiatric genetic studies: Is there an enrichment of 16p11.2 DEGs for the GWAS loci for ASD, SZ or ADHD? Can the authors also test for enrichment of postmortem disease signatures as reported by the CommonMind Consortium, NIH HBCC and/or PsychEncode UCLA (Geschwind) analyses?

We chose to focus on overlap between our identified DE gene list and the SFARI autism risk gene list, as 16p11.2 deletion is highly associated with ASD diagnosis and contains phenotypic overlap even in cases that do not meet the diagnostic criteria for ASD. We identified 8 genes in our DE gene list that have been identified by the SFARI Gene as genes with evidence for autism risk (Supplementary file 5).

Although this does not represent statistically significant enrichment, we might not expect to see enrichment of risk genes derived from SNP data at this timepoint. Established risk genes can potentially impact developmental trajectories at many timepoints throughout development, and we have focused our study on a single, early timepoint. Not every risk gene needs to be impactful at every timepoint for their cumulative effects on neurodevelopment to be manifest.

As an aside, it may be useful for the field to identify the specific timepoints at which various risk genes are impactful. We are heartened that our study pinpoints a limited set of 16p11.2 region genes and ASD-associated genes (e.g. AGMO, MUC4, EXT1), as this helps narrow the focus of what gene programs could potentially affect neurodevelopment at this stage.

b) Are gene expression differences validated in existing 16p11.2 postmortem, hiPSC or mouse datasets (ie. PMID: 26829649)?

We validated a select set of DE genes via qRT-PCR when our research group had access to our laboratory environment. Additionally, the downregulation of transcripts from the 16p11.2 deletion region is a phenomenon that has been observed in several other papers^1,2^.

One of our concerns is that comparison of our dataset to 16p11.2 postmortem or murine tissue would be impeded by the confound of timepoint. Our model represents a very early timepoint in fetal neurodevelopment which is strongly divergent from adult murine tissue or adult human postmortem brain.

Regarding comparison with other hiPSC models, we agree that this is an appropriate comparison to make, but a thorough meta-analysis of all available transcriptomics data is beyond the scope of the current manuscript.

c) Given some recent discussions of convergence between CNV effects in psychiatric disease, are major findings from 16p11.2 conserved in hiPSC neuron datasets from other major CNVs (ie. 22q11.2 hiPSC PMID: 27846841 and mouse PMID: 29666363; 15q11.2 BioRxiv https://doi.org/10.1101/772541; NRXN1 PMID: 31784728)

This is an interesting question which will merit further investigation but, as stated above, a meaningful comparison to these other datasets may not be possible due to the confound of timepoint and substantial differences in the intrinsic biologies of rosette-stage cortical neural progenitors and those of more differentiated cell types.

d) Can the authors discuss whether they believe the genome-wide transcriptomic effects of 16p11.2 reflect chromatin loops between the CNV locus and other target genes? miR targets? Or downstream impacts of perturbed neuronal maturation/activity/signaling?

We agree that this is a compelling concept. Chromosome looping is observed within the CNV region of 16p11.2^3^. The promoters of MVP and MAPK3 are both involved in longrange interactions with PTEN and CHD1L. MAPK3, but not MVP, was detected as a differentially expressed gene in our dataset (Figure 5F). CHD1L was detected in our dataset, but not differentially expressed. It is tempting to speculate that this long-range mechanism might help explain the appearance of PTENP1-AS, which regulates PTEN transcription, in the pink4 module associated with clinical phenotypes in DEL samples (Figure 6).

While loop calling and a thorough characterization of chromatin state is outside the scope of this study, we agree that this would be a fascinating area for follow-up.

7) Are the authors able to comment on any phenotypic differences that they did observe between microdeletion lines and wt lines?

Substantial efforts were made throughout the lifetime of this project to identify phenotypic differences between microdeletion and the wild-type lines.

For example, we were very interested in determining if the previously reported alterations in neural progenitor proliferation observed in mouse models was also detected in iPSC-derived progenitors.

Decreased expression of MAPK3, a key component of the Ras/ MAPK pathway implicated in cellular proliferation, and KCTD13 expression, another 16p11.2 interval gene, has reciprocal effects on brain size^5^. We thoroughly examined the mitotic index, growth rates, or rate of differentiation of the NPCs at various time points throughout the differentiation protocol. The only differences observed were driven entirely by the cryptic integration of Oct3/4 expressing reprogramming vectors.

In addition to evaluating the proliferative capacity of 16p11.2 NPCs, we characterized the differentiation capacity, neural rosette formation capacity, neural rosette size, degree of polarization, maturation speed, and cell cycle dynamics. None of these quantitative measures differed between microdeletion and wild-type hiPSC-derived NPCs.

We are in the process of analyzing single cell transcriptomics in the fetal cortex of 16p11.2 syntenic mice generated by Alea Mills and our preliminary clustering suggests that there is no difference in the relative abundance of cortical neural progenitors in DEL vs WT mouse fetal cortex. We understand that this is at odds with prior anatomical studies based on antibody staining of mouse cortical tissues and our hope is that the single cell transcriptomics in the mouse, along with the lack of proliferative differences in 16p11.2 DEL iPSC-derived progenitor cells may lead to a better understanding of the earlier data.

Revisions expected in follow-up work:

1) Characterization of micro-duplication lines.

We are currently in communication with the Simons Foundation to obtain and characterize additional 16p11.2 duplication lines given the relative dearth of integration-free duplication clones.

2) Elaboration of phenotypic differences at the neural progenitor stage or in differentiated neurons.

We are currently performing single nuclei RNAseq experiments with mature neurons from 16p11.2 deletion lines and are pleased by some encouraging results. We are hopeful that the in vitro experiments required to thoroughly validate these findings will be able to be performed in the near future.

**References**

Blumenthal, I. *et al.* Transcriptional consequences of 16p11.2 deletion and duplication in mouse cortex and multiplex autism families. *Am J Hum Genet* 94, 870-883, doi:10.1016/j.ajhg.2014.05.004 (2014).

Ward, T. R. *et al.* Genome-wide molecular effects of the neuropsychiatric 16p11 CNVs in an iPSCto-iN neuronal model. *bioRxiv*, 2020.2002.2009.940965, doi:10.1101/2020.02.09.940965 (2020).

Poot, M. Syndromes Hidden within the 16p11.2 Deletion Region. *Mol Syndromol* 9, 171-174, doi:10.1159/000490845 (2018).

Kusenda, M. *et al.* The Influence of Microdeletions and Microduplications of 16p11.2 on Global Transcription Profiles. *J Child Neurol* 30, 1947-1953, doi:10.1177/0883073815602066 (2015).

Golzio, C. *et al.* KCTD13 is a major driver of mirrored neuroanatomical phenotypes of the 16p11.2 copy number variant. *Nature* 485, 363-367, doi:10.1038/nature11091 (2012).